# Variational Speculative Decoding: Rethinking Draft Training from Token Likelihood to Sequence Acceptance

Xiandong Zou [1]   Jianshu Li [2]   Jing Huang [2]   Pan Zhou [1]

## Abstract

Speculative decoding accelerates inference for (M)LLMs, yet a training-decoding discrepancy persists: while existing methods optimize single greedy trajectories, decoding involves verifying and ranking multiple sampled draft paths. We propose *Variational Speculative Decoding* (VSD), formulating draft training as variational inference over latent proposals (draft paths). VSD maximizes the marginal probability of target-model acceptance, yielding an ELBO that promotes high-quality latent proposals while minimizing divergence from the target distribution. To enhance quality and reduce variance, we incorporate a path-level utility and optimize via an Expectation-Maximization procedure. The E-step draws Monte Carlo samples from an oracle-filtered posterior, while the M-step maximizes weighted likelihood using Adaptive Rejection Weighting (ARW) and Confidence-Aware Regularization (CAR). Theoretical analysis confirms that VSD increases expected acceptance length and speedup. Extensive experiments across LLMs and MLLMs show that VSD achieves up to a 9.6% speedup over EAGLE-3 and 7.9% over ViSpec, significantly improving decoding efficiency.

## 1. Introduction

Large language models (LLMs) such as GPT (Brown et al., 2020; Achiam et al., 2023) and LLaMA (Touvron et al., 2023a;b; Dubey et al., 2024), as well as multimodal LLMs (MLLMs) like LLaVA (Liu et al., 2024b), have achieved remarkable success in dialogue (Zheng et al., 2023; Goyal et al., 2017), coding (Chen, 2021), and reasoning (Cobbe et al., 2021; Kembhavi et al., 2016; Masry et al., 2022; Singh et al., 2019). Yet, most modern LLMs and MLLMs rely on

autoregressive decoding (Leviathan et al., 2023; Chen et al., 2023): each token must be generated sequentially based on all previous tokens, which limits parallelism and yields high latency and low throughput. Speculative decoding mitigates this by introducing a lightweight draft model to propose multiple tokens, which the target LLM or MLLM verifies in parallel. As a result, the target model can accept multiple tokens in a single forward pass without degrading the quality of generation, where longer accepted spans can equivalently translate into better inference speedup.

Recent work has improved speculative decoding by refining draft model training. For instance, HASS (Zhang et al., 2024) enforces feature consistency to reduce hidden-state mismatches, GRIFFIN (Hu et al., 2025b) resolves token-level misalignment, and EAGLE-3 (Li et al., 2026) incorporates training-time rollouts to better mimic decoding. However, a fundamental limitation remains: a *training-decoding distributional discrepancy*. The draft model is trained to favor a deterministic distribution, i.e., a single greedy path, while decoding operates over a stochastic distribution induced by ranked multi-path sampling, degrading the effectiveness of training for improving decoding performance.

To formalize this misalignment, we first consider the standard formulation of draft training. Given a context, the draft model is typically trained with cross-entropy to predict the next token generated by target model (Li et al., 2024a;b), which implicitly treats drafting as single-path prediction. The corresponding optimal policy is therefore *greedy and almost deterministic*: at each step, it concentrates probability mass on one most-likely continuation, producing a single "best" draft path. However, practical speculative decoding rarely uses a single greedy draft. Instead, it proposes multiple draft candidates via stochastic multi-path sampling—often in a tree (Li et al., 2024b; Hu et al., 2025b;a; Zhang et al., 2024)—and then *ranks* these candidates by path-level scores (e.g., cumulative draft probability) before sending only the top-$k$ tokens to the target model for verification. Thus, what matters at inference is not whether the greedy path is token-wise optimal, but whether the draft model favors a distribution which assigns high probability to the *set of draft candidates that survive ranking and are likely to be accepted by the target over longer horizons.*

---

[1]Singapore Management University [2]Ant Group. Correspondence to: Pan Zhou <panzhou@smu.edu.sg>.

*Proceedings of the 43$^{rd}$ International Conference on Machine Learning*, Seoul, South Korea. PMLR 306, 2026. Copyright 2026 by the author(s).

This training–decoding distributional discrepancy induced by two concrete misalignment. **(i) Deterministic training vs. stochastic decoding.** Standard cross-entropy training collapses the draft distribution toward a single greedy path, whereas practical decoding explicitly values maintaining a distribution in which its sampled several high-confidence draft paths that align with target-model preference and can pass its verification. **(ii) Token-level likelihood vs. path-level utility.** Training optimizes token-wise likelihood along one greedy trajectory given by the target model, but decoding makes decisions at the path level: draft candidates compete within a draft tree and are selected by cumulative confidence and relative ranking among siblings. As a result, a draft path that is locally optimal under token-level training can be pruned, while a sibling branch with slightly lower per-token likelihood may dominate in cumulative score and yield longer accepted spans. This creates a structural inefficiency: training expends capacity improving a greedy path that may not be used by the decoding algorithm.

We empirically verify the misalignment using an EAGLE-3 draft model with LLaMA-3.1-8B. As shown in Fig. 1(a), roughly 30% of training-time greedy paths are pruned during draft-tree construction, and the final accepted path coincides with the greedy path in only 36% of cases. Moreover, even when the greedy path is accepted, its average accepted length is only 3–4 tokens, compared to 5–6 tokens for alternative high-confidence candidates (Fig. 1(b)). These findings suggest that greedy, cross-entropy training optimizes the *wrong* draft distribution: it fails to prioritize the draft distribution favored by the decoding procedure and, ultimately, by the target model's acceptance behavior, limiting the achievable efficiency gains at inference time.

**Contributions.** To resolve the training–decoding distributional discrepancy in speculative decoding, we propose *Variational Speculative Decoding* (VSD), a principled training framework that directly targets the *distribution over draft paths that can be accepted by the target model*. Our contributions are highlighted below.

First, we propose a novel Variational Speculative Decoding (VSD) framework that reformulates draft model training as a variational inference problem. Unlike existing methods that rely on token-level supervision along one path, we treat the draft path as a latent proposal and aim to maximize the marginal likelihood of the target model's acceptance. Based on this formulation, we derive a principled Evidence Lower Bound (ELBO) that serves as our training objective. The ELBO includes two terms: the first encourages the generation of high-quality and acceptable latent proposals, and the second that acts as a regularizer to keep the draft policy aligned with the target distribution induced by the target model, effectively bridging the training-decoding gap.

Then, to optimize the intractable ELBO, we develop a tai-

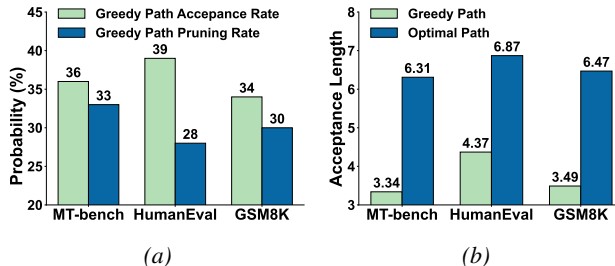

*Figure 1.* (a) Fraction where the accepted path coincides the greedy path and fraction of training-time greedy paths that are pruned during draft tree construction. (b) Comparison of acceptance length between greedy path and optimal high-confidence path.

lored EM algorithm. In the E-step, we approximate the variational posterior with oracle-filtered Monte Carlo sampling to concentrate on high-acceptance paths. In the M-step, we update the draft model by maximizing a weighted likelihood over these posterior samples. We further improve stability and efficiency with three components: *path-level utility* to favor longer accepted spans and reduce variance, *adaptive rejection weighting* to leverage informative rejected proposals, and *confidence-aware regularization* to discourage overconfident yet invalid paths that tend to fail verification.

Besides, we provide a rigorous theoretical analysis to justify our variational objective. We prove that maximizing the VSD objective is equivalent to increasing the lower bound of the *expected acceptance length*. By establishing a direct mathematical link between our variational bound and the wall-clock speedup ratio, we demonstrate that VSD is theoretically guaranteed to improve the efficiency of speculative decoding compared to traditional token-level likelihood-based training, given the fixed target distribution.

Finally, VSD consistently improves both acceptance length and inference speedup across language and multimodal settings. On MT-Bench (Zheng et al., 2023), HumanEval (Chen, 2021), and GSM8K (Cobbe et al., 2021) with LLaMA-3.1-8B, LLaMA-3.3-70B, DeepSeek-R1-Distill-LLaMA-8B, and Vicuna-13B, VSD outperforms EAGLE-3 (Li et al., 2026), improving acceptance length by 7.6% and speedup by 9.6%. On six multimodal benchmarks including SQA Image (Lu et al., 2022) and VQAv2 (Goyal et al., 2017) with LLaVA-1.5-7B/13B, VSD improves acceptance length by 7.9% and speedup by 7.3% over the best ViSpec baseline (Kang et al., 2025).

## 2. Related Work

Speculative decoding accelerates LLM inference by decoupling generation into a lightweight *drafting* stage and a parallel *verification* stage (Leviathan et al., 2023; Chen et al., 2023; Sun et al., 2023; Miao et al., 2024; Kim et al., 2023). To optimize the drafting stage, research has branched into several directions: removing separate drafters via auxiliary

heads (Cai et al., 2024) or layer-skipping (Hooper et al., 2025; Elhoushi et al., 2024; Liu et al., 2024a; Xia et al., 2024), and enhancing draft quality through token distillation (Zhou et al., 2023), N-gram-based prediction (Ou et al., 2024; Liu et al., 2025), or retrieval-augmented generation (He et al., 2024; Zhao et al., 2024).

On the verification side, modern methods have shifted from single-path to multi-path exploration to maximize throughput (Sun et al., 2023; Chen et al., 2024; Miao et al., 2024). This is exemplified by tree-based verification strategies like EAGLE (Li et al., 2024a;b) and EAGLE-3 (Li et al., 2026), which utilize uncertainty estimation and multi-token prediction to construct dynamic draft trees. Alternative paradigms, such as Jacobi decoding (Santilli et al., 2023; Teng et al., 2025) and Lookahead decoding (Fu et al., 2024), further accelerate inference by reformulating generation as a parallelizable optimization problem.

Despite these advancements, a fundamental *training-decoding distributional discrepancy* persists. While modern decoders sample, rank, and verify multiple draft paths (Li et al., 2026; Zhang et al., 2024; Hu et al., 2025b), the underlying draft models are still trained to concentrate probability mass on a single deterministic trajectory via token-level likelihood optimization. This mismatch forces the drafter to learn a distribution that diverges from the multi-path trajectories explored at inference time, ultimately limiting acceptance length and achievable speedups. To bridge this gap, we propose VSD framework, which reformulates draft training as a variational inference problem. By optimizing an MC-based EM framework (Moon, 1996; Gilks et al., 1995), VSD aligns the draft policy with the posterior distribution induced by multi-path decoding. VSD serves as a principled, complementary framework that enhances the efficiency of existing speculative decoding techniques.

## 3. Preliminary

Speculative decoding accelerates autoregressive generation via a *draft–verify* scheme. In draft phase, given the observed context (prefix) $\mathbf{x}$, the draft model $D$ generates the draft path (trajectory) $\mathbf{z} \sim q_\psi(\mathbf{z}|\mathbf{x})$, where $q_\psi(\mathbf{z}|\mathbf{x})$ denotes the induced distribution of $D$. Then in verification phase, the target model $T$ accepts each token $z_i$ if

$$\alpha_i(\mathbf{x}, \mathbf{z}_{<i}) = \min(1, p_\theta(\mathbf{z_i}|\mathbf{x}, \mathbf{z}_{<i})/q_\psi(\mathbf{z_i}|\mathbf{x}, \mathbf{z}_{<i})) \geq r, \tag{1}$$

where $\mathbf{z}_{<i}$ denotes tokens $\{\mathbf{z}_1, \mathbf{z}_2, \cdots, \mathbf{z}_{i-1}\}$, $p_\theta(\cdot|\cdot)$ is the induced distribution of target model $T$, and $r$ is sampled from a uniform distribution, i.e., $r \sim U[0,1]$. If $\mathbf{z_i}$ is rejected, the corresponding trajectory is truncated and a correction token is sampled from the distribution $\mathrm{norm}\big(\max\big(0, \ p_\theta(\cdot|\mathbf{x}, \mathbf{z}_{<i}) - q_\psi(\cdot|\mathbf{x}, \mathbf{z}_{<i})\big)\big)$, where $\mathrm{norm}$ denotes a normalization. This guarantees that the final output is an unbiased sample from $p_\theta(\cdot|\cdot)$.

## 4. Methodology

### 4.1. Variational Speculative Decoding

As introduced in Sec. 3, standard speculative decoding proceeds in a *draft–verify* manner: given a prefix $\mathbf{x}$, the draft model proposes a draft path $\mathbf{z}$, and the target model verifies it. A draft contributes to speedup only through the accepted prefix—the more tokens that survive verification, the greater the wall-clock speedup. Therefore, an effective training objective should directly favor draft proposals with a higher acceptance rate (ideally yielding long accepted spans), rather than merely matching the next-token distribution along a single greedy trajectory. This motivates addressing the training–decoding distributional discrepancy caused by (i) deterministic single-path training vs. stochastic multi-path decoding, and (ii) token-level likelihood vs. path-level utility.

To solve these two issues, we formulate the speculative decoding from a variational inference perspective below.

**Path-level Validity Probability.** Given a prefix $\mathbf{x}$, let $\mathbf{z} = (\mathbf{z}_1, \ldots, \mathbf{z}_\ell)$ be a draft path proposed by the draft model $q_\psi$. During verification, for each position $i$, the target model $p_\theta$ accepts the proposed token $z_i$ with probability $\alpha_i(\mathbf{x}, \mathbf{z}_{<i}) = \min\Big(1, \frac{p_\theta(z_i|\mathbf{x}, \mathbf{z}_{<i})}{q_\psi(z_i|\mathbf{x}, \mathbf{z}_{<i})}\Big)$, i.e., a sample $r \sim \mathrm{Unif}[0,1]$ is drawn and the token is accepted if $r \leq \alpha_i(\mathbf{x}, \mathbf{z}_{<i})$ (see Sec. 3). Accordingly, we define the *path-level validity probability* as the joint probability

$$\kappa(\mathbf{x}, \mathbf{z}) := \prod_{i=1}^{\ell} \alpha_i(\mathbf{x}, \mathbf{z}_{<i}) \in [0, 1], \tag{2}$$

which is the cumulative probability that all proposed tokens in $\mathbf{z}$ are accepted. Intuitively, $\kappa(\mathbf{x}, \mathbf{z})$ is an overall probability induced by the target model: paths with larger $\kappa$ are more likely to survive verification and thus yield greater speedup.

**A Latent-variable View of Verification.** We now model the verification as a probabilistic model. Let $\rho \in \{0, 1\}$ indicate whether a proposed draft path is valid (i.e., all its tokens are accepted). Conditioned on $\mathbf{x}$, define

$$\mathbf{z} \sim p_\theta(\mathbf{z} \mid \mathbf{x}), \qquad (\mathbf{z} \sim \text{prior distribution}) \tag{3}$$

$$\rho \mid (\mathbf{x}, \mathbf{z}) \sim \mathrm{Bernoulli}\big(\kappa(\mathbf{x}, \mathbf{z})\big). \tag{4}$$

In actual decoding, draft path $\mathbf{z}$ is sampled from the draft model: $\mathbf{z} \sim q_\psi(\mathbf{z} \mid \mathbf{x})$. In a variational derivation, we often define a prior reference distribution over latent proposal $\mathbf{z}$, like the Gaussian distribution as the prior in VAE (Kingma & Welling, 2013). $\mathbf{z} \sim p_\theta(\mathbf{z} \mid \mathbf{x})$ does not to describe how decoding samples $\mathbf{z}$, but to define a reference distribution over paths that we would like draft model $q_\psi$ to approximate.

Under this model, the marginal probability that a randomly drawn path is valid under verification equals

$$Z_\theta(\mathbf{x}) := p_\theta(\rho = 1 \mid \mathbf{x}) = \sum_{\mathbf{z}} p_\theta(\mathbf{z} \mid \mathbf{x}) \, \kappa(\mathbf{x}, \mathbf{z}). \quad (5)$$

Maximizing $\log Z_\theta(\mathbf{x})$ concentrates probability mass on the distribution of paths that are likely to pass verification, directly reflecting the decoding objective: the draft model should propose paths that the target model will accept.

**Variational Inference and an ELBO on** $\log Z_\theta(\mathbf{x})$**.** By Bayes' rule, the true posterior is

$$p_\theta(\mathbf{z} \mid \mathbf{x}, \rho = 1) = \frac{p_\theta(\mathbf{z} \mid \mathbf{x}) \, \kappa(\mathbf{x}, \mathbf{z})}{Z_\theta(\mathbf{x})} \; \propto \; p_\theta(\mathbf{z} \mid \mathbf{x}) \, \kappa(\mathbf{x}, \mathbf{z}),$$
$$(6)$$

which focuses on paths that are simultaneously plausible under the target model and likely to be accepted. However, the posterior (6) is intractable, as it requires summing over all possible latent proposals. In addition, during decoding, latent proposals must be generated by the draft model $q_\psi(\mathbf{z} \mid \mathbf{x})$ without accessing to the true posterior.

To make learning feasible, we introduce a variational distribution $q_\psi(\mathbf{z} \mid \mathbf{x})$ given by the draft model to derive an evidence lower bound (ELBO) (Kingma & Welling, 2013):

$$\log Z_\theta(\mathbf{x}) = \log \sum_{\mathbf{z}} q_\psi(\mathbf{z} \mid \mathbf{x}) \frac{p_\theta(\mathbf{z} \mid \mathbf{x}) \kappa(\mathbf{x}, \mathbf{z})}{q_\psi(\mathbf{z} \mid \mathbf{x})}$$

$$= \log \mathbb{E}_{\mathbf{z} \sim q_\psi(\cdot \mid \mathbf{x})} \left[ \frac{p_\theta(\mathbf{z} \mid \mathbf{x}) \kappa(\mathbf{x}, \mathbf{z})}{q_\psi(\mathbf{z} \mid \mathbf{x})} \right]$$

$$\geq \mathbb{E}_{\mathbf{z} \sim q_\psi(\cdot \mid \mathbf{x})} \Big[ \log \kappa(\mathbf{x}, \mathbf{z}) + \log p_\theta(\mathbf{z} \mid \mathbf{x}) - \log q_\psi(\mathbf{z} \mid \mathbf{x}) \Big]$$

$$= \underbrace{\mathbb{E}_{q_\psi(\mathbf{z} \mid \mathbf{x})}[\log \kappa(\mathbf{x}, \mathbf{z})]}_{\text{Prefer target-accepted paths}} - \underbrace{\mathbb{D}_{\mathrm{KL}}\big(q_\psi(\mathbf{z} \mid \mathbf{x}) \,\|\, p_\theta(\mathbf{z} \mid \mathbf{x})\big)}_{\text{Stay close to target distribution}}$$

$$=: \mathcal{L}_{\mathrm{VSD}}(\psi; \mathbf{x}), \quad (7)$$

where $\mathbb{D}_{\mathrm{KL}}(\cdot \mid \cdot)$ is the KL divergence and $\log \kappa(\mathbf{x}, \mathbf{z})$ denotes the log probability of proposing a valid draft path given the context. The ELBO admits a interpretation based on two terms: the draft model is encouraged to assign probability mass to paths with high acceptance probability, while remaining close to the reference distribution $p_\theta(\mathbf{z} \mid \mathbf{x})$ to avoid degenerate solutions. We yield our training objective:

$$\max_\psi \; \mathcal{L}_{\mathrm{VSD}}(\psi; \mathbf{x}). \quad (8)$$

This variational objective (8) resolves the two sources of training-decoding misalignment. (i) Deterministic vs. stochastic misalignment: by optimizing an expectation over $\mathbf{z} \sim q_\psi(\cdot \mid \mathbf{x})$, VSD explicitly trains under a *stochastic path distribution*, matching multi-path decoding rather than a single greedy trajectory. (ii) Token-level vs. path-level misalignment: $\log \kappa(\mathbf{x}, \mathbf{z})$ evaluates proposals at the *path level*, pushing probability mass toward trajectories that survive ranking/verification and typically yield longer accepted

spans. This differs from standard draft training that is locally optimal under token-level likelihood.

Finally, since exact computation of the posterior over valid paths is intractable, we optimize (8) using a stochastic Expectation-Maximization (EM) framework with Monte Carlo (MC) sampling: the E-step generates oracle-filtered path samples aligned with decoding-time exploration, and the M-step updates $q_\psi$ toward these posterior samples while controlling variance and avoiding overconfident collapse. Due to the space constraint, detailed algorithm is provided in Appendix B, with a high-level derivation in Sec. 4.2–4.3.

## 4.2. Expectation step

The E-step constructs an empirical approximation of the variational posterior over valid latent drafts, rather than committing to a single greedy decoding path. This is achieved via a stochastic Monte Carlo proposal-and-filter mechanism (Gilks et al., 1995). Specifically, the draft policy serves as a proposal distribution, generating a batch of candidates that are assessed by a path-level validity oracle (Algorithm 2 in Appendix B.1). This oracle acts as the constraint on the posterior support, retaining high-utility proposals within the valid support while rejecting low-quality ones. Rejected proposals are substituted with corrective samples drawn from the target policy, aligning the empirical distribution with the true posterior. This process yields a set of latent proposals approximating the support of the variational posterior distribution, capturing the multi-path nature of decoding.

## 4.3. Maximization Step

The M-step updates the draft model parameters $\psi$ by maximizing the log-likelihood of latent proposals sampled in the E-step. Formally, we estimate the gradient of the underlying variational objective to encourage the draft model to align with the posterior distribution of valid trajectories.

A straightforward estimator (Eqn. (20) in Appendix B.1) performs a standard maximum-likelihood update over proposed trajectories. However, directly optimizing the path-level posterior introduces two practical challenges: (i) high-variance gradients due to rejected latent proposals, and (ii) overconfident assignment of probability mass to locally plausible but globally invalid paths. Both issues lead to unstable training and inefficient draft distributions.

To address these challenges, we adopt a variance-reduced gradient estimator (Eqn. (23) in Appendix B.1), and incorporate the following complementary mechanisms: *Adaptive Rejection Weighting (ARW)* and *Confidence-Aware Regularization (CAR)*. They stabilize optimization and improve path-level confidence assignment.

**Adaptive Rejection Weighting (ARW).** To mitigate the gradient variance arising from rejected proposals, ARW acts

as a dynamic reweighting scheme inspired by variance reduction techniques in MC sampling (Gilks et al., 1995; Hoffman et al., 2023). Since standard gradients ignore rejected samples, ARW introduces a control variate coefficient $\beta$ (Eqn. (21) in Appendix B.1) that modulates the contribution of rejected samples based on the draft model's empirical reliability. When the draft model is weak ($\beta \approx 0$), most proposals are rejected and negative gradients are down-weighted, preventing unstable updates. When the draft model is strong ($\beta \approx 1$), rejections receive greater emphasis, improving discrimination among competing paths. By adapting the updated gradient to the draft model's reliability, ARW reduces gradient variance while preserving informative signals.

**Confidence-Aware Regularization (CAR).** While ARW stabilizes the update, it does not explicitly penalize the draft model for assigning high confidence to invalid paths—a critical issue that increases verification cost in speculative decoding (Li et al., 2024b). We address this issue by reweighting rejected proposals according to the draft model's confidence via $\zeta$ (Eqn. (22) in Appendix B.1), inspired by hard-negative mining (Schroff et al., 2015; Zou et al., 2025; Oh Song et al., 2016). Rejected paths assigned a high probability by the draft model receive larger penalties, while low-confidence rejections are down-weighted to preserve exploration. This confidence-aware reweighting encourages probability mass to concentrate on leading branches, yielding a more compact and effective draft tree. As a result, both draft path construction and verification overhead are reduced.

### 4.4. Theoretical Analysis

We first provide a theoretical analysis linking the proposed VSD objective to decoding-time efficiency. Conditioned on a prefix $\mathbf{x}$, the draft model proposes the latent proposal $\mathbf{z} = (\mathbf{z}_1, \mathbf{z}_2, \ldots, \mathbf{z}_\ell)$. Speculative verification proceeds token-by-token with acceptance probabilities $\alpha_i(\mathbf{x}, \mathbf{z}_{<i}) = \min\left(1, \frac{p_\theta(z_i|\mathbf{x}, \mathbf{z}_{<i})}{q_\psi(z_i|\mathbf{x}, \mathbf{z}_{<i})}\right)$ and we define the *survival probability* of the first $k$ tokens as

$$S_k(\mathbf{x}, \mathbf{z}) := \prod_{i=1}^{k} \alpha_i(\mathbf{x}, \mathbf{z}_{<i}), \qquad k = 1, \ldots, \ell. \quad (9)$$

In particular, the probability that the full length-$\ell$ path is accepted is $\kappa(\mathbf{x}, \mathbf{z}) := S_\ell(\mathbf{x}, \mathbf{z})$ (see Eqn. (2)).

**Expected Accepted Length.** Let $A(\mathbf{x}, \mathbf{z})$ be the random variable denoting the number of consecutive draft tokens accepted by the verifier, truncated at length $\ell$. Conditioned on $(\mathbf{x}, \mathbf{z})$, the event $\{A(\mathbf{x}, \mathbf{z}) \geq k\}$ denotes the first $k$ tokens are accepted, occurring with probability $S_k(\mathbf{x}, \mathbf{z})$. We have

$$\mathbb{E}\big[A(\mathbf{x}, \mathbf{z}) | \mathbf{x}, \mathbf{z}\big] = \sum_{k=1}^{\ell} \mathbb{P}\big(A(\mathbf{x}, \mathbf{z}) \geq k | \mathbf{x}, \mathbf{z}\big) = \sum_{k=1}^{\ell} S_k(\mathbf{x}, \mathbf{z}). \quad (10)$$

Given the latent proposal distribution $q_\psi(\mathbf{z} \mid \mathbf{x})$, we

define the *expected accepted length* as $\mathcal{A}(q_\psi; \mathbf{x}) = \mathbb{E}_{\mathbf{z} \sim q_\psi(\cdot|\mathbf{x})} \mathbb{E}\big[A(\mathbf{x}, \mathbf{z}) \mid \mathbf{x}, \mathbf{z}\big]$, and express it as

$$\mathcal{A}(q_\psi; \mathbf{x}) = \mathbb{E}_{\mathbf{z} \sim q_\psi(\cdot|\mathbf{x})}\Big[ \sum_{k=1}^{\ell} S_k(\mathbf{x}, \mathbf{z})\Big]. \quad (11)$$

**Theorem 1.** *By optimizing the VSD objective in Eqn. (8), the expected accepted length satisfies*

$$\mathcal{A}(q_\psi; \mathbf{x}) \geq \ell \exp\big(\mathcal{L}_{\mathrm{VSD}}(q_\psi; \mathbf{x})\big). \quad (12)$$

See its proof in Appendix C.1. Theorem 1 shows that increasing the VSD objective by a factor of $\Delta > 0$ increases a *provable lower bound* on the expected accepted length by a multiplicative factor of $e^\Delta$. As decoding speedup grows monotonically with the number of accepted draft tokens, this justifies optimizing $\mathcal{L}_{\mathrm{VSD}}$ for improved inference efficiency.

**Advantage over KL-only Draft Training.** We show that the VSD objective provides a strictly tighter lower bound on expected accepted length than KL-only draft training.

**Theorem 2.** *Let $q_\psi^\star(\cdot \mid \mathbf{x}) \in \arg\max_{q_\psi} \mathcal{L}_{\mathrm{VSD}}(q_\psi; \mathbf{x})$ be the optimizer of the VSD objective in Eqn. (8), and let $q_{\mathrm{KL}}(\cdot \mid \mathbf{x}) = p_\theta(\cdot \mid \mathbf{x})$ be the minimizer of the KL-only objective $\min_{q_\psi} \mathbb{D}_{\mathrm{KL}}(q_\psi \| p_\theta)$. Then,*

$$\begin{aligned} \mathcal{L}_{\mathrm{VSD}}(q_\psi^\star; \mathbf{x}) &= \log Z_\theta(\mathbf{x}) \\ &\geq \mathbb{E}_{\mathbf{z} \sim p_\theta(\cdot|\mathbf{x})}[\log \kappa(\mathbf{x}, \mathbf{z})] \quad (13) \\ &= \mathcal{L}_{\mathrm{VSD}}(q_{\mathrm{KL}}; \mathbf{x}), \end{aligned}$$

*with strict inequality whenever $\kappa(\mathbf{x}, \mathbf{z})$ is non-constant on the support of $p_\theta(\cdot \mid \mathbf{x})$. Consequently, the corresponding lower bounds on expected accepted length satisfy*

$$\ell \exp\Big(\mathcal{L}_{\mathrm{VSD}}(q_\psi^\star; \mathbf{x})\Big) \geq \ell \exp\Big(\mathcal{L}_{\mathrm{VSD}}(q_{\mathrm{KL}}; \mathbf{x})\Big). \quad (14)$$

The proof is provided in Appendix C.2. Theorem 2 establishes that VSD is strictly more effective than conventional KL-only draft training whenever different draft paths have non-uniform verification probabilities. Standard draft training minimizes only $\mathbb{D}_{\mathrm{KL}}(q_\psi \| p_\theta)$, which merely aligns the draft model with the target distribution and ignores whether sampled paths are likely to survive verification. In contrast, VSD additionally optimizes $\mathbb{E}_{q_\psi}[\log \kappa(\mathbf{x}, \mathbf{z})]$, explicitly favoring paths with higher verification validity probability. As a result, VSD achieves a strictly tighter lower bound on the expected accepted length, which is related to speculative decoding speedup. This formalizes the intuition that speculative decoding should prioritize both target-likely draft paths and draft paths that are more likely to survive verification.

**Step-level Variational Optimality in VSD.** We finally characterize the optimal draft distribution for each EM update in VSD. At the $t$-th EM update, the current draft model $q_{\psi^{(t)}}$ is used to compute the ELBO terms in the E-step. These terms are treated as fixed when optimizing the candidate draft distribution $q_\psi$ in the M-step.

**Theorem 3.** *Fix a prefix* $\mathbf{x}$ *and consider the* $t$-*th EM update. Let* $q_{\psi^{(t)}}(\cdot \mid \mathbf{x})$ *denote the current draft distribution used in the E-step. Define the step-level path validity probability*

$$\kappa_t(\mathbf{x}, \mathbf{z}) = \prod_{i=1}^{\ell} \min\left(1, \frac{p_\theta(z_i \mid \mathbf{x}, \mathbf{z}_{<i})}{q_{\psi^{(t)}}(z_i \mid \mathbf{x}, \mathbf{z}_{<i})}\right). \quad (15)$$

*Then the corresponding step-level valid-path posterior is*

$$p_{\theta,t}(\mathbf{z} \mid \mathbf{x}, \rho = 1) = \frac{p_\theta(\mathbf{z} \mid \mathbf{x}) \, \kappa_t(\mathbf{x}, \mathbf{z})}{Z_{\theta,t}(\mathbf{x})}, \quad (16)$$

*where*

$$Z_{\theta,t}(\mathbf{x}) = \sum_{\mathbf{z}} p_\theta(\mathbf{z} \mid \mathbf{x}) \kappa_t(\mathbf{x}, \mathbf{z}). \quad (17)$$

*For any candidate distribution* $q_\psi(\cdot \mid \mathbf{x})$ *over draft paths, which serves as the optimization variable in the M-step, define the step-level VSD surrogate*

$$\begin{aligned} \mathcal{L}_{\mathrm{VSD}}^{(t)}(q_\psi; \mathbf{x}) = \;& \mathbb{E}_{\mathbf{z} \sim q_\psi(\cdot \mid \mathbf{x})}[\log \kappa_t(\mathbf{x}, \mathbf{z})] \\ & - \mathbb{D}_{\mathrm{KL}}(q_\psi(\cdot \mid \mathbf{x}) \| p_\theta(\cdot \mid \mathbf{x})). \end{aligned} \quad (18)$$

*We have the unique maximizer of* $\mathcal{L}_{\mathrm{VSD}}^{(t)}(q_\psi; \mathbf{x})$ *is*

$$q_t^\star(\cdot \mid \mathbf{x}) = p_{\theta,t}(\cdot \mid \mathbf{x}, \rho = 1). \quad (19)$$

The proof is provided in Appendix C.3. Theorem 3 gives a step-level variational characterization of VSD. At the $t$-th EM update, the ELBO terms induced by the current draft model $q_{\psi^{(t)}}$ are fixed during the M-step. Therefore, the posterior $p_{\theta,t}(\mathbf{z} \mid \mathbf{x}, \rho = 1)$ is a fixed step-level target, and the M-step surrogate has a well-defined global optimum. This result shows that VSD can be viewed as an iterative procedure that repeatedly constructs a verification-aware posterior and updates the draft model toward this posterior. However, KL-only draft training (Zhang et al., 2024; Hu et al., 2025b; Li et al., 2026) only aligns the draft model with the unconditional target distribution $p_\theta(\mathbf{z} \mid \mathbf{x})$ without explicitly accounting for whether draft paths are likely to survive during the verification.

## 5. Experiment

**Models & Tasks.** We evaluate our method across a diverse set of LLMs, including LLaMA-3.1-Instruct-8B, LLaMA-3.3-Instruct-70B (Dubey et al., 2024), Vicuna-1.3-13B (Chiang et al., 2023), and DeepSeek-R1-Distill-LLaMA-8B (Guo et al., 2025). These experiments are conducted on a single RTX PRO 6000 GPU, with the exception of the LLaMA-3.3-70B variant, which requires two GPUs. Performance is assessed on three widely adopted benchmarks: MT-Bench (Zheng et al., 2023) for multi-turn dialogue, HumanEval (Chen, 2021) for code generation, and GSM8K (Cobbe et al., 2021) for mathematical reasoning. Furthermore, we extend our evaluation to MLLMs, assessing VSD

on the open-source LLaVA-1.5 family (7B and 13B) (Liu et al., 2024b). These experiments are conducted on an L-40S GPU across six diverse multimodal benchmarks, including visual question answering (SQA Image (Lu et al., 2022), VQAv2 (Goyal et al., 2017), AI2D (Kembhavi et al., 2016)), document/chart understanding (ChartQA (Masry et al., 2022), TextVQA (Singh et al., 2019)), and hallucination evaluation (Hallusion (Guan et al., 2024)).

**Metrics.** We follow priors (Li et al., 2026; Hu et al., 2025b), and test all methods using a decoding batch size of 1 under temperatures $T \in \{0, 1\}$. Since VSD preserves the output distribution of the target model and is lossless, we focus on efficiency metrics: (i) **Speedup Ratio** ($SR$), defined as the wall-clock time acceleration relative to vanilla decoding, and (ii) **Acceptance Length** ($\tau$), representing the average number of tokens accepted per draft-verification cycle.

**Baselines.** Vanilla autoregressive decoding is used as the baseline (speedup ratio = 1.00×). For comparison, we include recent state-of-the-art speculative decoding methods: SPS (Leviathan et al., 2023), PLD (Saxena, 2023), Lookahead (Fu et al., 2024), Medusa (Cai et al., 2024), EAGLE (Li et al., 2024a), EAGLE-2 (Li et al., 2024b), HASS (Zhang et al., 2024), GRIFFIN (Hu et al., 2025b), EAGLE-3 (Li et al., 2026), MSD (Lin et al., 2025), and ViSpec (Kang et al., 2025). For all baselines, their official models are evaluated on the same hardware (RTX PRO 6000 or L-40S) for benchmarking. Note that our measured $\tau$ values for these baselines closely align with their officially reported results.

**Implementation.** For fairness, when integrated with baseline (e.g., EAGLE-3, MSD), VSD augments the original training objective by adding the first term in Eqn. (7) to train the draft model. The original training loss (e.g., standard cross-entropy (Li et al., 2026) or top-$k$ KL loss (Zhang et al., 2024)) can be seen as the approximation to the KL term in the variational objective. VSD does not change other factors like the size or network design of draft model. By default, VSD initializes its draft model from the one provided by EAGLE-3, MSD, and ViSpec. To assess compatibility, we also experiment with draft models trained by other approaches (see Tab. 3). For LLM, we train DeepSeek-R1-Distill-LLaMA-8B on the OpenThoughts-114k-math dataset (Guha et al., 2025), while the remaining draft models are trained using the ShareGPT dataset (Chiang et al., 2023) following prior work. For MLLM, we train all draft models with ShareGPT (Chiang et al., 2023) and LLaVA-Instruct-150K (Liu et al., 2024b). Detailed training configurations are provided in Appendices F.1 and F.2.

### 5.1. Main Results

**Results on LLMs.** As shown in Tab. 1, VSD consistently outperforms all baselines, including the state-of-the-art EAGLE-3, across all datasets, models, and temperature

*Table 1.* Comparison of speedup ratio ($SR$) and acceptance length ($\tau$) on standard LLM benchmarks with temperature $T \in \{0, 1\}$. The subscripts denote the relative improvement compared to the corresponding baseline. For example, at $T = 0$, EAGLE-3+VSD on LLaMA-3.1 Instruct 8B achieves the average $SR$ of 4.22 with an additional +6.7% gain over the EAGLE-3 value of 3.95.

| Model | Method | Temperature = 0 | | | | | | | | Temperature = 1 | | | | | | | |
|---|---|---|---|---|---|---|---|---|---|---|---|---|---|---|---|---|---|
| | | MT-bench | | HumanEval | | GSM8K | | Average | | MT-bench | | HumanEval | | GSM8K | | Average | |
| | | $SR\uparrow$ | $\tau\uparrow$ | $SR\uparrow$ | $\tau\uparrow$ | $SR\uparrow$ | $\tau\uparrow$ | $SR\uparrow$ | $\tau\uparrow$ | $SR\uparrow$ | $\tau\uparrow$ | $SR\uparrow$ | $\tau\uparrow$ | $SR\uparrow$ | $\tau\uparrow$ | $SR\uparrow$ | $\tau\uparrow$ |
| **LLaMA-3.1 Instruct 8B** | PLD | 1.69 | 1.71 | 1.90 | 1.80 | 1.94 | 1.97 | 1.84 | 1.83 | N/A, since the acceptance conditions are relaxed | | | | | | | |
| | Lookahead | 1.82 | 1.80 | 2.03 | 1.88 | 2.05 | 2.06 | 1.97 | 1.91 | | | | | | | | |
| | EAGLE | 2.03 | 3.26 | 2.84 | 3.49 | 2.27 | 3.30 | 2.38 | 3.35 | 1.62 | 2.18 | 2.23 | 3.15 | 2.08 | 2.74 | 1.98 | 2.69 |
| | EAGLE-2 | 2.93 | 4.37 | 3.88 | 5.06 | 3.14 | 4.53 | 3.32 | 4.65 | 2.01 | 2.83 | 2.99 | 4.49 | 2.56 | 3.51 | 2.52 | 3.61 |
| | GRIFFIN | 3.41 | 5.10 | 4.35 | 6.32 | 3.48 | 5.55 | 3.75 | 5.66 | 2.23 | 3.53 | 3.67 | 5.52 | 2.83 | 4.41 | 2.91 | 4.48 |
| | EAGLE-3 | 3.79 | 6.31 | 4.29 | 6.87 | 3.77 | 6.47 | 3.95 | 6.55 | 2.32 | 4.10 | 3.48 | 5.86 | 3.10 | 5.07 | 2.97 | 5.01 |
| | **EAGLE-3+VSD** | **4.05** | **6.79** | **4.63** | **7.27** | **3.96** | **6.93** | **4.22**$_{+6.7\%}$ | **7.00**$_{+6.8\%}$ | **2.41** | **4.23** | **3.80** | **6.15** | **3.31** | **5.39** | **3.18**$_{+7.1\%}$ | **5.26**$_{+4.9\%}$ |
| **Vicuna-1.3 13B** | SPS | 1.42 | 2.29 | 1.58 | 2.53 | 1.25 | 1.95 | 1.42 | 2.26 | 1.21 | 1.85 | 1.29 | 1.96 | 1.05 | 1.77 | 1.18 | 1.86 |
| | Hydra | 1.91 | 3.32 | 2.30 | 3.78 | 1.96 | 3.42 | 2.06 | 3.51 | N/A, since the acceptance conditions are relaxed | | | | | | | |
| | EAGLE | 2.11 | 3.77 | 2.44 | 4.21 | 2.03 | 3.65 | 2.19 | 3.87 | 1.65 | 3.15 | 1.90 | 3.52 | 1.69 | 3.45 | 1.75 | 3.37 |
| | EAGLE-2 | 2.81 | 4.88 | 3.52 | 5.45 | 2.81 | 4.73 | 3.05 | 5.02 | 2.65 | 4.43 | 2.91 | 4.92 | 2.23 | 4.43 | 2.60 | 4.59 |
| | EAGLE-3 | 3.59 | 6.73 | 4.22 | 7.47 | 3.59 | 6.52 | 3.80 | 6.90 | 3.08 | 5.77 | 3.51 | 6.45 | 2.98 | 5.92 | 3.19 | 6.04 |
| | **EAGLE-3+VSD** | **4.08** | **7.11** | **4.74** | **8.10** | **4.02** | **7.09** | **4.28**$_{+12.6\%}$ | **7.43**$_{+7.7\%}$ | **3.26** | **6.08** | **3.73** | **6.74** | **3.26** | **6.23** | **3.42**$_{+7.1\%}$ | **6.35**$_{+5.0\%}$ |
| **DeepSeek-R1 Distill-LLaMA 8B** | PLD | 1.41 | 1.48 | 1.72 | 1.71 | 1.66 | 1.65 | 1.60 | 1.61 | N/A, since the acceptance conditions are relaxed | | | | | | | |
| | Lookahead | 1.64 | 1.71 | 1.89 | 1.83 | 1.87 | 1.84 | 1.80 | 1.79 | | | | | | | | |
| | EAGLE-2 | 2.23 | 4.02 | 3.05 | 4.35 | 3.02 | 4.50 | 2.77 | 4.29 | 2.23 | 3.65 | 2.66 | 3.95 | 2.69 | 4.39 | 2.53 | 3.99 |
| | GRIFFIN | 2.89 | 4.44 | 3.62 | 5.57 | 3.89 | 6.03 | 3.47 | 5.35 | 2.42 | 4.06 | 2.99 | 4.78 | 3.57 | 5.71 | 2.99 | 4.85 |
| | EAGLE-3 | 3.56 | 5.57 | 4.09 | 6.26 | 4.31 | 6.72 | 3.99 | 6.18 | 2.77 | 4.70 | 3.34 | 5.24 | 4.01 | 6.40 | 3.37 | 5.44 |
| | **EAGLE-3+VSD** | **4.07** | **5.93** | **4.67** | **6.92** | **4.61** | **7.29** | **4.45**$_{+11.6\%}$ | **6.71**$_{+8.6\%}$ | **3.03** | **5.11** | **3.84** | **5.63** | **4.18** | **6.79** | **3.68**$_{+9.2\%}$ | **5.84**$_{+7.4\%}$ |
| **LLaMA-3.3 Instruct 70B** | PLD | 1.37 | 1.59 | 1.53 | 1.74 | 1.51 | 1.72 | 1.47 | 1.68 | N/A, since the acceptance conditions are relaxed | | | | | | | |
| | Lookahead | 1.48 | 1.73 | 1.65 | 1.85 | 1.71 | 1.92 | 1.61 | 1.83 | | | | | | | | |
| | EAGLE-2 | 2.97 | 3.92 | 3.72 | 4.59 | 3.17 | 4.30 | 3.29 | 4.27 | 3.18 | 3.98 | 3.39 | 4.48 | 2.97 | 4.25 | 3.18 | 4.24 |
| | EAGLE-3 | 3.56 | 5.62 | 4.22 | 6.48 | 3.94 | 6.28 | 3.91 | 6.13 | 3.52 | 5.47 | 3.87 | 6.16 | 3.73 | 5.81 | 3.71 | 5.82 |
| | **EAGLE-3+VSD** | **3.81** | **6.04** | **4.51** | **6.94** | **4.27** | **6.71** | **4.20**$_{+7.5\%}$ | **6.56**$_{+7.1\%}$ | **3.73** | **5.81** | **4.14** | **6.67** | **3.88** | **6.33** | **3.92**$_{+5.7\%}$ | **6.27**$_{+7.8\%}$ |

*Table 2.* Comparison of speedup ratio $SR$ and acceptance length $\tau$ on standard MLLM benchmarks with temperature $T \in \{0, 1\}$. The subscripts denote the relative improvement compared to the corresponding baseline. For example, at $T = 0$, MSD+VSD on LLaVA-1.5 7B achieves the average $SR$ of 2.45 with an additional +7.9% gain over the MSD value of 2.27.

| | Model | Method | VQAv2 | | AI2D | | SQA Image | | ChartQA | | TextVQA | | Hallusion | | Average | |
|---|---|---|---|---|---|---|---|---|---|---|---|---|---|---|---|---|
| | | | $SR\uparrow$ | $\tau\uparrow$ | $SR\uparrow$ | $\tau\uparrow$ | $SR\uparrow$ | $\tau\uparrow$ | $SR\uparrow$ | $\tau\uparrow$ | $SR\uparrow$ | $\tau\uparrow$ | $SR\uparrow$ | $\tau\uparrow$ | $SR\uparrow$ | $\tau\uparrow$ |
| **Temperature = 0** | **LLaVA-1.5 7B** | Lookahead | 1.43 | 1.35 | 1.60 | 1.46 | 1.51 | 1.45 | 1.33 | 1.43 | 1.32 | 1.31 | 1.59 | 1.51 | 1.46 | 1.42 |
| | | Medusa | 1.63 | 1.52 | 1.71 | 1.69 | 1.59 | 1.57 | 1.50 | 1.62 | 1.49 | 1.51 | 1.74 | 1.99 | 1.61 | 1.65 |
| | | EAGLE-2 | 2.37 | 4.53 | 1.98 | 3.35 | 1.79 | 3.41 | 1.81 | 3.25 | 1.84 | 3.62 | 2.04 | 3.55 | 1.97 | 3.62 |
| | | MSD | 2.47 | 4.99 | 2.23 | 4.20 | 2.08 | 4.24 | 2.21 | 4.14 | 2.12 | 4.18 | 2.51 | 4.60 | 2.27 | 4.40 |
| | | **MSD+VSD** | **2.69** | **5.10** | **2.39** | **4.48** | **2.22** | **4.53** | **2.46** | **4.58** | **2.29** | **4.41** | **2.67** | **4.93** | **2.45**$_{+7.9\%}$ | **4.67**$_{+6.3\%}$ |
| | | ViSpec | 2.57 | 5.08 | 2.30 | 4.36 | 2.19 | 4.44 | 2.32 | 4.30 | 2.25 | 4.31 | 2.61 | 4.73 | 2.37 | 4.54 |
| | | **ViSpec+VSD** | **2.78** | **5.19** | **2.47** | **4.69** | **2.32** | **4.70** | **2.54** | **4.69** | **2.40** | **4.52** | **2.77** | **5.03** | **2.54**$_{+7.2\%}$ | **4.80**$_{+5.9\%}$ |
| | **LLaVA-1.5 13B** | Lookahead | 1.42 | 1.30 | 1.32 | 1.40 | 1.32 | 1.41 | 1.66 | 1.46 | 1.38 | 1.33 | 1.43 | 1.47 | 1.42 | 1.40 |
| | | Medusa | 1.50 | 1.41 | 1.47 | 1.61 | 1.39 | 1.50 | 1.75 | 1.56 | 1.51 | 1.58 | 1.62 | 2.01 | 1.54 | 1.61 |
| | | EAGLE-2 | 2.36 | 4.29 | 2.04 | 3.38 | 1.76 | 3.16 | 2.21 | 3.18 | 1.81 | 3.35 | 2.09 | 3.41 | 2.05 | 3.46 |
| | | MSD | 2.44 | 4.48 | 2.35 | 4.08 | 2.10 | 3.93 | 2.87 | 3.97 | 2.25 | 4.07 | 2.53 | 4.35 | 2.43 | 4.15 |
| | | **MSD+VSD** | **2.65** | **4.95** | **2.60** | **4.50** | **2.30** | **4.56** | **3.21** | **4.50** | **2.49** | **4.54** | **2.77** | **4.79** | **2.68**$_{+10.1\%}$ | **4.64**$_{+11.9\%}$ |
| | | ViSpec | 2.55 | 4.66 | 2.43 | 4.22 | 2.21 | 4.29 | 3.01 | 4.24 | 2.34 | 4.21 | 2.63 | 4.51 | 2.53 | 4.36 |
| | | **ViSpec+VSD** | **2.73** | **5.07** | **2.66** | **4.61** | **2.35** | **4.62** | **3.29** | **4.60** | **2.55** | **4.66** | **2.89** | **4.87** | **2.74**$_{+8.5\%}$ | **4.74**$_{+8.8\%}$ |
| **Temperature = 1** | **LLaVA-1.5 7B** | Lookahead | 1.34 | 1.24 | 1.12 | 1.26 | 1.24 | 1.30 | 1.28 | 1.27 | 1.08 | 1.21 | 1.31 | 1.38 | 1.23 | 1.28 |
| | | Medusa | 1.51 | 1.93 | 1.19 | 1.43 | 1.38 | 1.87 | 1.36 | 1.82 | 1.12 | 1.43 | 1.47 | 1.88 | 1.34 | 1.73 |
| | | EAGLE-2 | 1.85 | 3.28 | 1.24 | 2.61 | 1.48 | 2.71 | 1.52 | 2.57 | 1.37 | 2.72 | 1.52 | 2.82 | 1.50 | 2.79 |
| | | MSD | 2.06 | 3.59 | 1.49 | 3.13 | 1.77 | 3.26 | 1.81 | 3.02 | 1.46 | 2.93 | 1.80 | 3.39 | 1.74 | 3.22 |
| | | **MSD+VSD** | **2.12** | **3.64** | **1.69** | **3.58** | **2.01** | **3.39** | **1.84** | **3.20** | **1.76** | **3.06** | **1.86** | **3.51** | **1.88**$_{+8.2\%}$ | **3.40**$_{+5.4\%}$ |
| | | ViSpec | 2.15 | 3.67 | 1.58 | 3.37 | 1.90 | 3.33 | 1.86 | 3.18 | 1.62 | 2.97 | 1.84 | 3.41 | 1.83 | 3.32 |
| | | **ViSpec+VSD** | **2.21** | **3.79** | **1.74** | **3.65** | **2.08** | **3.46** | **1.92** | **3.29** | **1.81** | **3.11** | **1.91** | **3.62** | **1.94**$_{+6.5\%}$ | **3.64**$_{+5.0\%}$ |
| | **LLaVA-1.5 13B** | Lookahead | 1.09 | 1.21 | 1.23 | 1.29 | 1.18 | 1.26 | 1.08 | 1.28 | 1.17 | 1.22 | 1.20 | 1.32 | 1.16 | 1.26 |
| | | Medusa | 1.16 | 1.35 | 1.31 | 1.43 | 1.25 | 1.39 | 1.16 | 1.87 | 1.21 | 1.94 | 1.35 | 1.42 | 1.24 | 1.57 |
| | | EAGLE-2 | 1.71 | 3.17 | 1.60 | 2.63 | 1.58 | 2.67 | 1.13 | 2.59 | 1.62 | 2.58 | 1.68 | 2.82 | 1.55 | 2.74 |
| | | MSD | 1.81 | 3.44 | 1.81 | 3.08 | 1.77 | 3.21 | 1.39 | 3.06 | 1.92 | 3.01 | 1.96 | 3.41 | 1.78 | 3.20 |
| | | **MSD+VSD** | **2.00** | **3.54** | **1.88** | **3.27** | **1.88** | **3.51** | **1.54** | **3.33** | **2.04** | **3.21** | **2.15** | **3.66** | **1.91**$_{+7.8\%}$ | **3.42**$_{+6.7\%}$ |
| | | ViSpec | 1.89 | 3.49 | 1.85 | 3.18 | 1.79 | 3.24 | 1.43 | 3.20 | 1.98 | 3.15 | 2.01 | 3.45 | 1.83 | 3.29 |
| | | **ViSpec+VSD** | **2.04** | **3.61** | **1.90** | **3.29** | **1.90** | **3.54** | **1.56** | **3.35** | **2.11** | **3.29** | **2.19** | **3.74** | **1.95**$_{+6.8\%}$ | **3.47**$_{+5.6\%}$ |

*Table 3.* Comparison of speedup ratio ($SR$) and acceptance length ($\tau$) when respectively using draft models trained by GRIFFIN and HASS as initialization of VSD on LLaMA-3.1 Instruct-8B.

| Method | Temperature = 0 | | | | | | | | Temperature = 1 | | | | | | | |
| | MT-bench | | HumanEval | | GSM8K | | Average | | MT-bench | | HumanEval | | GSM8K | | Average | |
| | $SR\uparrow$ | $\tau\uparrow$ | $SR\uparrow$ | $\tau\uparrow$ | $SR\uparrow$ | $\tau\uparrow$ | $SR\uparrow$ | $\tau\uparrow$ | $SR\uparrow$ | $\tau\uparrow$ | $SR\uparrow$ | $\tau\uparrow$ | $SR\uparrow$ | $\tau\uparrow$ | $SR\uparrow$ | $\tau\uparrow$ |
|---|---|---|---|---|---|---|---|---|---|---|---|---|---|---|---|---|
| GRIFFIN | 3.51 | 5.39 | 4.42 | 6.27 | 3.61 | 5.79 | 3.85 | 5.82 | 2.56 | 3.96 | 3.73 | 5.75 | 3.29 | 4.77 | 3.19 | 4.83 |
| **GRIFFIN+VSD** | **3.85** | **5.75** | **4.56** | **6.53** | **3.85** | **6.11** | **4.09** | **6.13** | **2.70** | **4.14** | **3.95** | **6.10** | **3.48** | **5.31** | **3.38** | **5.18** |
| HASS | 3.21 | 4.95 | 4.25 | 6.05 | 3.23 | 5.24 | 3.56 | 5.42 | 2.34 | 3.77 | 3.49 | 5.52 | 3.11 | 4.59 | 2.98 | 4.63 |
| **HASS+VSD** | **3.60** | **5.42** | **4.47** | **6.40** | **3.44** | **5.47** | **3.84** | **5.76** | **2.52** | **3.98** | **3.67** | **5.84** | **3.33** | **5.05** | **3.17** | **4.96** |

*Table 4.* Ablation study of adaptive rejection weighting (ARW), confidence-aware regularization (CAR), and number of latent proposals ($S$) in VSD. We report speedup ratio ($SR$) and acceptance length ($\tau$) on LLaMA-3-Instruct-8B with temperature $T \in \{0, 1\}$.

| Settings | | | Temperature = 0 | | | | | | | | Temperature = 1 | | | | | | | |
| ARW | CAR | $S$ | MT-bench | | HumanEval | | GSM8K | | Average | | MT-bench | | HumanEval | | GSM8K | | Average | |
| | | | $SR\uparrow$ | $\tau\uparrow$ | $SR\uparrow$ | $\tau\uparrow$ | $SR\uparrow$ | $\tau\uparrow$ | $SR\uparrow$ | $\tau\uparrow$ | $SR\uparrow$ | $\tau\uparrow$ | $SR\uparrow$ | $\tau\uparrow$ | $SR\uparrow$ | $\tau\uparrow$ | $SR\uparrow$ | $\tau\uparrow$ |
|---|---|---|---|---|---|---|---|---|---|---|---|---|---|---|---|---|---|---|
| - | - | 10 | 3.81 | 6.32 | 4.33 | 6.90 | 3.78 | 6.51 | 3.97 | 6.58 | 2.34 | 4.12 | 3.52 | 5.86 | 3.12 | 5.18 | 2.99 | 5.05 |
| ✓ | - | 10 | 3.88 | 6.43 | 4.41 | 7.04 | 3.82 | 6.56 | 4.05 | 6.68 | 2.34 | 4.12 | 3.55 | 5.91 | 3.16 | 5.21 | 3.02 | 5.08 |
| ✓ | ✓ | 10 | 3.93 | 6.52 | 4.49 | 7.17 | 3.86 | 6.63 | 4.07 | 6.77 | 2.36 | 4.14 | 3.57 | 5.94 | 3.19 | 5.23 | 3.04 | 5.10 |
| - | - | 20 | 3.85 | 6.42 | 4.39 | 6.99 | 3.83 | 6.56 | 4.02 | 6.66 | 2.36 | 4.15 | 3.63 | 6.06 | 3.19 | 5.24 | 3.06 | 5.15 |
| ✓ | - | 20 | 3.93 | 6.53 | 4.48 | 7.13 | 3.88 | 6.68 | 4.10 | 6.78 | 2.38 | 4.21 | 3.69 | 6.09 | 3.23 | 5.29 | 3.10 | 5.19 |
| ✓ | ✓ | 20 | 3.98 | 6.62 | 4.55 | 7.26 | 3.91 | 6.85 | 4.15 | 6.91 | 2.40 | 4.24 | 3.73 | 6.11 | 3.26 | 5.31 | 3.13 | 5.22 |
| - | - | 40 | 3.88 | 6.44 | 4.43 | 7.06 | 3.87 | 6.67 | 4.06 | 6.73 | 2.37 | 4.17 | 3.70 | 6.09 | 3.24 | 5.30 | 3.10 | 5.19 |
| ✓ | - | 40 | 3.99 | 6.64 | 4.54 | 7.17 | 3.92 | 6.84 | 4.15 | 6.88 | 2.39 | 4.18 | 3.76 | 6.12 | 3.28 | 5.34 | 3.14 | 5.21 |
| ✓ | ✓ | 40 | 4.05 | 6.79 | 4.63 | 7.27 | 3.96 | 6.93 | 4.22 | 7.00 | 2.41 | 4.23 | 3.80 | 6.15 | 3.31 | 5.39 | 3.18 | 5.26 |

*Table 5.* Ablation study on batch size using LLaMA-3.1 Instruct 8B. We report speedup ratio ($SR$) and acceptance length ($\tau$) on LLaMA-3-Instruct-8B with temperature $T \in \{0, 1\}$ under batch sizes 4 and 8.

| Batch Size | Method | Temperature = 0 | | | | | | | | Temperature = 1 | | | | | | | |
| | | MT-bench | | HumanEval | | GSM8K | | Average | | MT-bench | | HumanEval | | GSM8K | | Average | |
| | | $SR\uparrow$ | $\tau\uparrow$ | $SR\uparrow$ | $\tau\uparrow$ | $SR\uparrow$ | $\tau\uparrow$ | $SR\uparrow$ | $\tau\uparrow$ | $SR\uparrow$ | $\tau\uparrow$ | $SR\uparrow$ | $\tau\uparrow$ | $SR\uparrow$ | $\tau\uparrow$ | $SR\uparrow$ | $\tau\uparrow$ |
|---|---|---|---|---|---|---|---|---|---|---|---|---|---|---|---|---|---|
| **4** | EAGLE-3 | 2.44 | 4.65 | 2.82 | 5.42 | 2.64 | 4.81 | 2.63 | 4.96 | 2.04 | 3.12 | 2.58 | 4.71 | 2.30 | 3.89 | 2.30 | 3.91 |
| | **EAGLE-3+VSD** | **2.66** | **5.02** | **3.06** | **5.74** | **2.77** | **5.16** | **2.83** | **5.31** | **2.13** | **3.26** | **2.83** | **4.98** | **2.51** | **4.17** | **2.49** | **4.14** |
| **8** | EAGLE-3 | 1.54 | 4.68 | 1.80 | 5.44 | 1.68 | 4.80 | 1.67 | 4.97 | 1.25 | 3.22 | 1.57 | 4.78 | 1.44 | 3.96 | 1.42 | 3.99 |
| | **EAGLE-3+VSD** | **1.66** | **5.03** | **1.97** | **5.79** | **1.81** | **5.18** | **1.81** | **5.33** | **1.35** | **3.30** | **1.69** | **5.01** | **1.58** | **4.22** | **1.54** | **4.17** |

settings. On average, VSD accepts 6–7 tokens per drafting–verification cycle, exceeding the 5–6 tokens achieved by EAGLE-3. As a result, VSD improves EAGLE-3's wall-clock speedup by an average of 9.6% under greedy decoding ($T = 0$) and 7.3% under stochastic decoding ($T = 1$), while preserving the lossless property of speculative decoding.

The gains of VSD are closely related to the draft–target ga. When greedy training collapses to suboptimal paths that deviate from the target model's preferred trajectories, VSD brings larger improvements by optimizing the draft policy toward valid path-level posteriors. This trend is supported by our EAGLE-3 experiments: tasks and models with lower Greedy Path Acceptance Rate generally obtain larger gains from VSD. For example, on MT-Bench, Vicuna-1.3-13B achieves a 13.6% speedup-ratio improvement at $T = 0$, whereas LLaMA-3.1-70B obtains a smaller gain of 5.7%. This is consistent with their average Greedy Path Acceptance Rates: Vicuna-1.3-13B has a lower rate of 30%, while LLaMA-3.1-70B reaches 36%, indicating that the stronger model's greedy paths are already better aligned with target

trajectories and thus leave less room for improvement.

VSD also yields larger benefits on tasks requiring long-horizon consistency. Benchmarks with longer average outputs, such as MT-Bench and HumanEval, exhibit more pronounced improvements across model sizes. In code generation task, VSD increases the speedup ratio by 12.4% and 14.2% at $T = 0$ for LLaMA-3.1-8B and DeepSeek-R1-8B, respectively, highlighting its effectiveness for structured generation. On mathematical reasoning task (GSM8K), VSD also consistently surpasses EAGLE-3, with DeepSeek-R1-8B achieving gains of 7.0% at $T = 0$ and 4.2% at $T = 1$, suggesting that VSD better captures sequential reasoning patterns critical for problem solving.

Finally, VSD is more effective under greedy decoding. At $T = 0$, the target distribution is more concentrated, which sharpens the valid-path posterior and makes it easier to align the draft policy with this path-level posterior. At $T = 1$, the target distribution has higher entropy and accepts a wider range of drafts. As a result, the valid-path posterior becomes

more dispersed, making it harder to align the draft policy with it and thereby reducing the relative efficiency gain.

**Results on MLLMs.** As shown in Tab. 2, VSD significantly improves inference efficiency on standard MLLM benchmarks. When integrated with MSD and ViSpec, VSD consistently increases both speedup ratio ($SR$) and acceptance length ($\tau$) across model scales and decoding temperatures.

The improvements on MLLMs follow the same pattern observed in LLMs. VSD provides larger gains when the baseline draft paths are less aligned with the target model and when the task requires longer-horizon consistency. For deterministic greedy decoding ($T = 0$), VSD robustly generalizes to LLaVA-1.5-13B, improving MSD's average speedup and acceptance length by up to 10.1% and 11.9%, respectively, while improving ViSpec by 8.5% and 8.8%. Under stochastic decoding ($T = 1$), VSD maintains consistent superiority, increasing the average speedup of MSD by 7.8% and that of ViSpec by 6.8% on LLaVA-1.5-13B.

In addition, across diverse visual-language tasks, the largest improvements appear on benchmarks involving more structured or long-horizon reasoning, such as ChartQA and AI2D, while gains on short-horizon tasks are relatively smaller.

**Compatibility Evaluation.** To further test the compatibility and transferability of VSD, we apply VSD to finetune draft models of other speculative methods, i.e., GRIFFIN and HASS, on LLaMA-3-Instruct-8B. In Tab. 3, both GRIFFIN+VSD and HASS+VSD achieve consistent performance gains. For greedy decoding ($T = 0$), GRIFFIN+VSD improves average speedup and acceptance length by 6.3% and 5.4%, respectively, while HASS+VSD yields improvements of 7.6% and 6.4%. Similarly, in stochastic decoding ($T = 1$), VSD increases the $SR$ of GRIFFIN and HASS by 5.8% and 6.5%, respectively. These results on LLMs and the improvements of integrating VSD with MSD and ViSpec confirm VSD's versatility across distinct backbones and its effectiveness to align training and decoding objectives.

### 5.2. Ablation Study

**Effect of number of latent proposals $S$.** We first analyze the performance under varying numbers of latent proposals $S \in \{10, 20, 40\}$. As shown in Tab. 4, increasing $S$ consistently improves both the speedup ratio ($SR$) and the average acceptance length ($\tau$) across benchmarks. For example, at temperature $T = 0$, $SR$ increases from 3.97 to 4.06 and $\tau$ from 6.58 to 6.73 on average when $S$ grows from 10 to 40. This scaling phenomenon demonstrates that VSD can effectively leverage draft samples and bridge the objective misalignment between training and decoding.

**Effect of adaptive rejection weighting (ARW).** Incorporating ARW improves both $SR$ and $\tau$ across $S$ and temperature. Without ARW, sample efficiency is reduced, resulting

in weaker draft paths and shorter acceptance lengths. For instance, at $S = 40$ and $T = 0$, adding ARW increases the average $SR$ from 4.06 to 4.15 and $\tau$ from 6.73 to 6.88. This confirms that ARW effectively reduces variance in the Monte Carlo updates, leading to an efficient draft policy.

**Effect of confidence-aware regularization (CAR).** Adding CAR on top of ARW further stabilizes training and improves performance, since CAR mitigates the influence of overconfident hard samples. As a result, the average $SR$ and $\tau$ have additional gains; for example, at $S = 40$ and $T = 0$, the average $SR$ increases from 4.15 to 4.22.

**Ablation on Batch Size.** Tab. 5 studies the effect of batch size on LLaMA-3.1 Instruct 8B. At $T = 0$, VSD increases the average speedup ratio by 7.6% and 8.4% for batch sizes 4 and 8, respectively, while improving the average acceptance length by 7.1% and 7.2%. Similar trends are observed at $T = 1$, where VSD improves the average speedup ratio by 8.3% and 8.5% for batch sizes 4 and 8.

Although the speedup ratio naturally decreases when increasing the batch size from 4 to 8 due to the reduced benefit of speculative decoding under heavier parallel workloads, VSD maintains relative gains over EAGLE-3, confirming that VSD enhances the intrinsic quality of draft tokens.

Overall, these results validate the key design choices in VSD and highlight its robustness across benchmark tasks.

## 6. Conclusion

We introduce VSD, a principled framework that resolves the training–decoding distributional discrepancy in speculative decoding. VSD formulates draft training as variational inference over latent draft paths, directly favoring verification-valid proposals via path-level utility. An Monte Carlo-based EM algorithm enables efficient optimization. Theoretical analysis validates that VSD increases expected acceptance length and speedup. Experiments across LLMs and MLLMs show consistent improvements in acceptance length and state-of-the-art decoding speedups, complementary to existing speculative decoding methods.

**Limitations.** Due to computational constraints, it is hard for us to scale the number of latent proposals beyond $S = 40$ in Monte Carlo estimation during draft model training. Given the observed scaling behavior of VSD in Sec. 5.2, we expect that both speedup ratio and acceptance length would have further improvements with larger $S$. Moreover, our experiments mainly focus on text and visual tasks; extending VSD to other modalities like speech remains as future work.

## Acknowledgements

This work was supported by Ant Group, and the Singapore Ministry of Education (MOE) Academic Research Fund (AcRF) Tier 1 grant (Proposal ID: 25-SIS-SMU-003).

## Impact Statement

The pretrained LLMs and MLLMs employed in this research may reflect biases or generate sensitive or potentially offensive content, intended solely for academic and scientific purposes. The opinions expressed within generated outputs do not represent the views of the authors. We remain committed to fostering the development of AI technologies which align with ethical standards and reflect societal values.

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

# A. Notation and Definitions

The major notations used throughout the paper are summarized in Table 6.

*Table 6.* Major notations used in VSD.

| Notation | Definition |
|---|---|
| $\mathbf{x}$ | Input prefix/context. |
| $\mathbf{z} = (z_1, \ldots, z_\ell)$ | Draft path proposed by the draft model. |
| $\ell$ | Length of the draft path. |
| $q_\psi(\mathbf{z} \mid \mathbf{x})$ | Draft model distribution parameterized by $\psi$. |
| $p_\theta(\mathbf{z} \mid \mathbf{x})$ | Reference/target path distribution parameterized by $\theta$. |
| $\alpha_i(\mathbf{x}, \mathbf{z}_{<i})$ | Token-level acceptance probability at step $i$. |
| $\kappa(\mathbf{x}, \mathbf{z})$ | Path-level validity probability: $\prod_{i=1}^{\ell} \alpha_i(\mathbf{x}, \mathbf{z}_{<i})$. |
| $\rho \in \{0, 1\}$ | Indicator variable of whether a draft path is fully accepted. |
| $Z_\theta(\mathbf{x})$ | Marginal probability of generating a valid path: $p_\theta(\rho = 1 \mid \mathbf{x})$. |
| $p_\theta(\mathbf{z} \mid \mathbf{x}, \rho = 1)$ | Valid-path posterior distribution. |
| $\mathcal{L}_{\mathrm{VSD}}$ | Variational speculative decoding objective (ELBO). |
| $S_k(\mathbf{x}, \mathbf{z})$ | Survival probability of the first $k$ draft tokens. |
| $\mathcal{A}(q_\psi; \mathbf{x})$ | Expected accepted length under draft distribution $q_\psi$. |
| $A(\mathbf{x}, \mathbf{z})$ | Random variable representing accepted draft length. |
| $S$ | Number of latent proposals sampled in the E-step. |
| $\beta$ | Adaptive rejection weighting coefficient in ARW. |
| $\zeta$ | Confidence-aware regularization weight in CAR. |

# B. Algorithm

## B.1. Algorithm Detail

---

**Algorithm 1** MC-based EM Framework for Speculative Draft Training

---

**input** Prefix $\mathbf{x}$, draft policy $q_\psi$, target policy $p_\theta$, oracle $o$, sampling length $L$, sampling size $S$.
**output** Parameters $\psi$ of the draft model.
1: // E-step: Stochastic Latent Proposal
2: **for** $\ell = 1, \ldots, L$ **do**
3:     **for** $s = 1, \ldots, S$ **do**
4:         $\tilde{\mathbf{z}}_{<\ell}^{(s)} \sim q_\psi(\cdot \mid \mathbf{x})$
5:         **if** $o(\tilde{\mathbf{z}}_{<\ell}^{(s)}, \mathbf{x}) > 0$ **then**
6:             $\mathbf{z}_{<\ell}^{(s)} \leftarrow \tilde{\mathbf{z}}_{<\ell}^{(s)}$
7:         **else**
8:             $\mathbf{z}_{<\ell}^{\prime(s)} \sim p_\theta(\cdot \mid \mathbf{x})$
9:             $\mathbf{z}_{<\ell}^{(s)} \leftarrow \mathbf{z}_{<\ell}^{\prime(s)}$
10:         **end if**
11:     **end for**
12: **end for**
13: // M-step: Draft Policy Optimization
14: Let $\tilde{o}^{(\ell,s)} = o(\tilde{\mathbf{z}}_{<\ell}^{(s)}, \mathbf{x})$
15: Let $o^{\prime(\ell,s)} = o(\mathbf{z}_{<\ell}^{(s)}, \mathbf{x})$
16: Estimate gradient $\hat{g}$ via `basic_gradient`

$$\hat{g} = \frac{1}{\sum_{\ell=1}^{L} \sum_{s=1}^{S} o^{\prime(l,s)}} \sum_{\ell=1}^{L} \sum_{s=1}^{S} o^{\prime(l,s)} \nabla_\psi \log q_\psi(\mathbf{z}_{<\ell}^{(s)} \mid \mathbf{x}) \tag{20}$$

17: or `control_variate_gradient` with given temperature $\tau$

$$\beta^{(\ell,s)} \leftarrow \frac{\sum_{s' \neq s} o^{\prime(\ell,s')} \tilde{o}^{(\ell,s')}}{\sum_{s' \neq s} o^{\prime(\ell,s')}}, \tag{21}$$

$$\zeta^{(\ell,s)} \leftarrow \frac{\exp(q_\psi(\tilde{\mathbf{z}}_{<\ell}^{(s)} \mid \mathbf{x})/\tau)}{\sum_{j=1}^{S} \exp(q_\psi(\tilde{\mathbf{z}}_{<\ell}^{(j)} \mid \mathbf{x})/\tau)}. \tag{22}$$

$$\hat{g} = \frac{1}{\sum_{\ell=1}^{L} \sum_{s=1}^{S} o^{\prime(l,s)}} \sum_{\ell=1}^{L} \sum_{s=1}^{S} o^{\prime(l,s)} \Big[ \nabla_\psi \log q_\psi(\mathbf{z}_{<\ell}^{(s)} \mid \mathbf{x}) - \beta^{(l,s)} \cdot \zeta^{(l,s)} \cdot \nabla_\psi \log q_\psi(\tilde{\mathbf{z}}_{<\ell}^{(s)} \mid \mathbf{x}) \Big] \tag{23}$$

18: Update draft parameters: $\psi \leftarrow \text{Optimizer}(\psi, \hat{g})$

---

---

**Algorithm 2** Oracle $o(\mathbf{z}_{<\ell}, \mathbf{x})$

---

**input** Greedy target policy $p_\theta^{(1)}$, latent proposal $\mathbf{z}_{<\ell}$, prefix $\mathbf{x}$, indicator function $\mathbb{I}[\cdot]$, weighting factor $w(\ell)$.
**output** Scalar $o$.
1: // Greedy proposal baseline
2: $\bar{\mathbf{z}}_{<\ell} \sim p_\theta^{(1)}(\cdot \mid \mathbf{x})$
3: // Compute path-level utility based on $\kappa$ defined in Eqn. (2)
4: $o(\mathbf{z}_{<\ell}, \mathbf{x}) \leftarrow w(\ell) \mathbb{I}[\log \kappa(\mathbf{x}, \mathbf{z}_{<\ell}) \geq \log \kappa(\mathbf{x}, \bar{\mathbf{z}}_{<\ell})]$

---

## B.2. Algorithm Rationale

**Oracle.** The oracle (Algorithm 2) provides a principled mechanism for optimizing the path-level validity term in the ELBO. Rather than rewarding all sampled trajectories equally, it selects trajectories whose utility matches or exceeds that of a reference trajectory, thereby concentrating learning on paths that are more likely to survive verification. We use the greedy target path as the reference because it represents the standard upper bound in traditional distillation.

**Expectation Step.** The E-step (Algorithm 1) approximates the posterior distribution of valid speculative paths. Rather than relying solely on the static target distribution, we employ a *proposal-and-correction* scheme to construct a set of high path-level utility latent proposals.

For each sampling length $\ell \in [1, L]$, the draft policy $q_\psi$ first acts as a proposal distribution, generating $S$ latent proposals $\tilde{\mathbf{z}}_{<\ell}$. This stochastic proposal prevents the mode collapse often observed in greedy distillation and ensures exploration of the solution space. To enforce alignment with the target model, we introduce a verification oracle $o(\cdot)$ (Algorithm 2) that acts as a gating function. The oracle based on the indicator function $\mathbb{I}(\cdot)$ validates a latent proposal only if its path-level decoding utility $\log \kappa(\mathbf{x}, \mathbf{z}_{<\ell})$, where $\kappa$ defined in Eqn. (2), matches or exceeds that of a greedy baseline $\bar{\mathbf{z}}$ generated by the target model $p_\theta^{(1)}$.

Crucially, to maintain a consistent training signal, we do not simply discard rejected latent proposals. Instead, when a draft fails verification (i.e., $o(\cdot) = 0$), we use a correction mechanism by resampling a substitute path $\mathbf{z}'_{<\ell}$ from the target policy $p_\theta(\cdot \mid \mathbf{x})$. This ensures that the set of latent proposals $\{\mathbf{z}_{<\ell}^{(s)}\}$ constitutes a valid approximation of the posterior distribution supported by the target model. Finally, a length-dependent weight $w(\ell)$ is applied to assign higher variance reduction importance to longer, valid latent proposals (see Appendix F).

**Maximization step.** The M-step (Algorithm 1) updates the draft policy parameters $\psi$ to maximize the likelihood of the latent proposals identified in the E-step. We employ a variance-reduced gradient estimator that dynamically weighs the contribution of latent proposals. The gradient is in the form (see the exact form in Eqn. (23)):

$$\hat{g} = \mathbb{E}_{\ell,s}\left[ \underbrace{\nabla_\psi \log q_\psi(\mathbf{z}_{<\ell}^{(s)} \mid \mathbf{x})}_{\text{Target Proposal}} - \beta^{(\ell,s)} \zeta^{(\ell,s)} \underbrace{\nabla_\psi \log q_\psi(\tilde{\mathbf{z}}_{<\ell}^{(s)} \mid \mathbf{x})}_{\text{Proposal Correction}} \right]. \tag{24}$$

This estimator incorporates two critical mechanisms—i) Adaptive Rejection Weighting ($\beta$) and ii) Confidence-Aware Regularization ($\zeta$)—to mitigate the gradient variance.

*i) Adaptive Rejection Weighting (ARW).* To stabilize training across different phases of convergence, ARW introduces a control variate coefficient $\beta^{(\ell,s)}$ that estimates the reliability of the current draft policy. We compute $\beta$ using a leave-one-out estimator to minimize bias:

$$\beta^{(\ell,s)} = \frac{\sum_{j\neq s} o'^{(\ell,j)} \tilde{o}^{(\ell,j)}}{\sum_{j\neq s} o'^{(\ell,j)}}. \tag{25}$$

Here, $\tilde{o}$ represents the binary validity of the latent proposal, while $o'$ is the weight of the final corrected latent proposal. Mathematically, $\beta^{(\ell,s)}$ serves as a proxy for the *local acceptance rate*.

- *Low Reliability Regime ($\beta \to 0$):* When the draft model is weak, most proposals are rejected. ARW naturally anneals the "Proposal Correction" term, reducing the gradient to standard supervised learning on the target-corrected samples $\mathbf{z}$. This prevents noisy, high-variance rejection signals from destabilizing early training.

- *High Reliability Regime ($\beta \to 1$):* As the model converges, $\beta$ approaches 1. For valid proposals where $\mathbf{z} = \tilde{\mathbf{z}}$, the two gradient terms cancel out, effectively zeroing the loss for paths that are already optimal. This acts as an *automatic early stopping* mechanism for mastered proposals, focusing the model's capacity on difficult, rejected branches (where $\mathbf{z} \neq \tilde{\mathbf{z}}$).

It adaptively balances positive and negative signals. By conditioning gradient updates on path-level acceptance statistics, ARW reduces gradient variance while preserving informative rejection signals, guiding the draft policy toward the multi-path decoding distribution induced by the target model.

*ii) Confidence-Aware Regularization (CAR).* While ARW manages gradient variance, it treats all rejections equally. However, *confident errors*—where the model assigns high probability to invalid paths—are particularly detrimental to speculative tree efficiency (Li et al., 2024b).

To address this issue, we introduce Confidence-Aware Regularization, which reweights rejected latent proposals based on their confidence under the draft model. Specifically, rejected latent proposals $\tilde{\mathbf{z}}_{<\ell}^{(s)}$ are assigned weights

$$\zeta^{(\ell,s)} = \frac{\exp(q_\psi(\tilde{\mathbf{z}}_{<\ell}^{(s)} \mid \mathbf{x})/\tau)}{\sum_{j=1}^{S} \exp(q_\psi(\tilde{\mathbf{z}}_{<\ell}^{(j)} \mid \mathbf{x})/\tau)}, \tag{26}$$

where $\tau$ controls the strength of confidence-based reweighting (see detailed $\tau$ Appendix F).

This formulation functions as a *hard-negative mining* mechanism. By normalizing across the batch $S$, CAR penalizes relative overconfidence: a rejected latent proposal that is significantly more confident than its peers receives a dominating gradient penalty ($\zeta \uparrow$), reducing its likelihood. Conversely, low-confidence exploration errors are down-weighted, preserving the policy's entropy and preventing mode collapse.

# C. Proof

## C.1. Proof of Theorem 1

*Proof.* Recall the verification-centric variational objective given a fixed prefix $\mathbf{x}$:

$$\mathcal{L}_{\text{VSD}}(q_\psi; \mathbf{x}) := \mathbb{E}_{\mathbf{z} \sim q_\psi(\cdot \mid \mathbf{x})}\big[\log \kappa(\mathbf{x}, \mathbf{z})\big] - \mathbb{D}_{\text{KL}}\big(q_\psi(\cdot \mid \mathbf{x}) \,\|\, p_\theta(\cdot \mid \mathbf{x})\big), \tag{27}$$

where $p_\theta(\mathbf{z} \mid \mathbf{x})$ is a target model distribution.

Since $\alpha_i \in (0, 1]$, we have $S_1(\mathbf{x}, \mathbf{z}) \geq \cdots \geq S_\ell(\mathbf{x}, \mathbf{z}) = \kappa(\mathbf{x}, \mathbf{z})$. Therefore

$$\sum_{k=1}^{\ell} S_k(\mathbf{x}, \mathbf{z}) \geq \sum_{k=1}^{\ell} \kappa(\mathbf{x}, \mathbf{z}) = \ell\,\kappa(\mathbf{x}, \mathbf{z}). \tag{28}$$

Taking expectation over $\mathbf{z} \sim q_\psi(\cdot \mid \mathbf{x})$ yields

$$\mathcal{A}\big(q_\psi; \mathbf{x}\big) \geq \ell\, \mathbb{E}_{\mathbf{z} \sim q_\psi(\cdot \mid \mathbf{x})}\big[\kappa(\mathbf{x}, \mathbf{z})\big]. \tag{29}$$

By Jensen's inequality, for any $q_\psi(\mathbf{z} \mid \mathbf{x})$,

$$\mathbb{E}_{q_\psi}\big[\kappa(\mathbf{x}, \mathbf{z})\big] \geq \exp\Big(\mathbb{E}_{q_\psi}\big[\log \kappa(\mathbf{x}, \mathbf{z})\big]\Big). \tag{30}$$

Combining (29) and (30), we yield

$$\mathcal{A}\big(q_\psi; \mathbf{x}\big) \geq \ell\, \exp\Big(\mathbb{E}_{q_\psi}\big[\log \kappa(\mathbf{x}, \mathbf{z})\big]\Big). \tag{31}$$

Thus, any training procedure that increases $\mathbb{E}_{q_\psi}[\log \kappa(\mathbf{x}, \mathbf{z})]$ strictly increases the guaranteed lower bound on the expected accepted length.

Since $\mathbb{D}_{\text{KL}}(\cdot \| \cdot) \geq 0$, we have

$$\mathcal{L}_{\text{VSD}}(q_\psi; \mathbf{x}) \leq \mathbb{E}_{q_\psi}\big[\log \kappa(\mathbf{x}, \mathbf{z})\big]. \tag{32}$$

Combining (31) and (32), we have

$$\mathcal{A}\big(q_\psi; \mathbf{x}\big) \geq \ell\, \exp\Big(\mathcal{L}_{\text{VSD}}(q_\psi; \mathbf{x})\Big). \tag{33}$$

Therefore, if training increases the VSD loss by $\Delta > 0$ on the same prefix $\mathbf{x}$ (or in expectation over prefixes), then the guaranteed lower bound on expected accepted length increases multiplicatively by a factor $e^\Delta$. $\square$

## C.2. Proof of Theorem 2

*Proof.* By definition,

$$Z_\theta(\mathbf{x}) = \mathbb{E}_{\mathbf{z} \sim p_\theta(\cdot | \mathbf{x})}[\kappa(\mathbf{x}, \mathbf{z})]. \tag{34}$$

Since $\log(\cdot)$ is concave, Jensen's inequality gives

$$\begin{aligned}
\log Z_\theta(\mathbf{x}) &= \log \mathbb{E}_{\mathbf{z} \sim p_\theta(\cdot | \mathbf{x})}[\kappa(\mathbf{x}, \mathbf{z})] \\
&\geq \mathbb{E}_{\mathbf{z} \sim p_\theta(\cdot | \mathbf{x})}[\log \kappa(\mathbf{x}, \mathbf{z})].
\end{aligned} \tag{35}$$

The inequality is strict whenever $\kappa(\mathbf{x}, \mathbf{z})$ is not almost surely constant under $p_\theta(\cdot \mid \mathbf{x})$.

For the KL-only baseline, the minimizer of $\mathbb{D}_{\mathrm{KL}}(q_\psi \| p_\theta)$ is $q_{\mathrm{KL}}(\cdot \mid \mathbf{x}) = p_\theta(\cdot \mid \mathbf{x})$. Therefore,

$$\begin{aligned}
\mathcal{L}_{\mathrm{VSD}}(q_{\mathrm{KL}}; \mathbf{x}) &= \mathbb{E}_{\mathbf{z} \sim p_\theta(\cdot | \mathbf{x})}[\log \kappa(\mathbf{x}, \mathbf{z})] - \mathbb{D}_{\mathrm{KL}}(p_\theta(\cdot \mid \mathbf{x}) \,\|\, p_\theta(\cdot \mid \mathbf{x})) \\
&= \mathbb{E}_{\mathbf{z} \sim p_\theta(\cdot | \mathbf{x})}[\log \kappa(\mathbf{x}, \mathbf{z})].
\end{aligned} \tag{36}$$

On the other hand, since $q_\psi^\star$ maximizes the VSD objective, and the ELBO is tight at the variational optimum, we have

$$\mathcal{L}_{\mathrm{VSD}}(q_\psi^\star; \mathbf{x}) = \log Z_\theta(\mathbf{x}). \tag{37}$$

Combining this equality with Jensen's inequality yields

$$\mathcal{L}_{\mathrm{VSD}}(q_\psi^\star; \mathbf{x}) = \log Z_\theta(\mathbf{x}) \geq \mathcal{L}_{\mathrm{VSD}}(q_{\mathrm{KL}}; \mathbf{x}), \tag{38}$$

with strict inequality whenever $\kappa(\mathbf{x}, \mathbf{z})$ is non-constant on the support of $p_\theta(\cdot \mid \mathbf{x})$.

Finally, applying the accepted-length lower bound in Eqn. (33) to both $q_\psi^\star$ and $q_{\mathrm{KL}}$, and using the monotonicity of the exponential function, we obtain

$$\ell \exp\big(\mathcal{L}_{\mathrm{VSD}}(q_\psi^\star; \mathbf{x})\big) \geq \ell \exp(\mathcal{L}_{\mathrm{VSD}}(q_{\mathrm{KL}}; \mathbf{x})). \tag{39}$$

$\square$

## C.3. Proof of Theorem 3

We first provide a formal statement of Theorem 3. At the $t$-th EM update, the current draft model $q_{\psi^{(t)}}$ is used to compute the ELBO terms in the E-step. These terms are treated as fixed when optimizing the candidate draft distribution $q_\psi$ in the M-step.

**Theorem.** *Fix a prefix $\mathbf{x}$ and consider the $t$-th EM update. Let $q_{\psi^{(t)}}(\cdot \mid \mathbf{x})$ be the current draft distribution used in the E-step, and assume it has full support on the draft path space. Let $p_\theta(\mathbf{z} \mid \mathbf{x})$ be the target reference distribution over draft paths. Define the step-level path validity probability*

$$\kappa_t(\mathbf{x}, \mathbf{z}) = \prod_{i=1}^\ell \min\left(1, \frac{p_\theta(z_i \mid \mathbf{x}, \mathbf{z}_{<i})}{q_{\psi^{(t)}}(z_i \mid \mathbf{x}, \mathbf{z}_{<i})}\right), \tag{40}$$

*and assume $\kappa_t(\mathbf{x}, \mathbf{z}) > 0$ on the support considered. Define*

$$Z_{\theta, t}(\mathbf{x}) := \sum_{\mathbf{z}} p_\theta(\mathbf{z} \mid \mathbf{x}) \kappa_t(\mathbf{x}, \mathbf{z}) = \mathbb{E}_{\mathbf{z} \sim p_\theta(\cdot | \mathbf{x})}[\kappa_t(\mathbf{x}, \mathbf{z})], \tag{41}$$

*and the corresponding step-level valid-path posterior*

$$p_{\theta, t}(\mathbf{z} \mid \mathbf{x}, \rho = 1) := \frac{p_\theta(\mathbf{z} \mid \mathbf{x}) \, \kappa_t(\mathbf{x}, \mathbf{z})}{Z_{\theta, t}(\mathbf{x})}. \tag{42}$$

*For any candidate distribution $q_\psi(\cdot \mid \mathbf{x})$ over draft paths, which serves as the optimization variable in the M-step, define the step-level VSD surrogate*

$$\mathcal{L}_{\mathrm{VSD}}^{(t)}(q_\psi; \mathbf{x}) := \mathbb{E}_{\mathbf{z} \sim q_\psi(\cdot | \mathbf{x})}[\log \kappa_t(\mathbf{x}, \mathbf{z})] - \mathbb{D}_{\mathrm{KL}}(q_\psi(\cdot \mid \mathbf{x}) \,\|\, p_\theta(\cdot \mid \mathbf{x})). \tag{43}$$

*Then:*

1. (**Variational identity**) *For any such* $q_\psi(\cdot \mid \mathbf{x})$,

$$\mathcal{L}_{\mathrm{VSD}}^{(t)}(q_\psi; \mathbf{x}) = \log Z_{\theta,t}(\mathbf{x}) - \mathbb{D}_{\mathrm{KL}}(q_\psi(\cdot \mid \mathbf{x}) \,\|\, p_{\theta,t}(\cdot \mid \mathbf{x}, \rho = 1)). \tag{44}$$

2. (**Optimality for the step-level surrogate**) *We have*

$$\max_{q_\psi} \mathcal{L}_{\mathrm{VSD}}^{(t)}(q_\psi; \mathbf{x}) = \log Z_{\theta,t}(\mathbf{x}), \tag{45}$$

*and the unique maximizer is*

$$q_t^\star(\cdot \mid \mathbf{x}) = p_{\theta,t}(\cdot \mid \mathbf{x}, \rho = 1). \tag{46}$$

*Proof.* We first prove the variational identity. The optimality claims then follow from the non-negativity of KL divergence. By definition of KL divergence,

$$\begin{aligned} &\mathbb{D}_{\mathrm{KL}}(q_\psi(\cdot \mid \mathbf{x}) \,\|\, p_{\theta,t}(\cdot \mid \mathbf{x}, \rho = 1)) \\ &= \sum_{\mathbf{z}} q_\psi(\mathbf{z} \mid \mathbf{x}) \log \frac{q_\psi(\mathbf{z} \mid \mathbf{x})}{p_{\theta,t}(\mathbf{z} \mid \mathbf{x}, \rho = 1)}. \end{aligned} \tag{47}$$

Substituting the posterior definition in Eqn. (42), we have

$$\log p_{\theta,t}(\mathbf{z} \mid \mathbf{x}, \rho = 1) = \log p_\theta(\mathbf{z} \mid \mathbf{x}) + \log \kappa_t(\mathbf{x}, \mathbf{z}) - \log Z_{\theta,t}(\mathbf{x}). \tag{48}$$

Plugging Eqn. (48) into Eqn. (47) yields

$$\begin{aligned} &\mathbb{D}_{\mathrm{KL}}(q_\psi \,\|\, p_{\theta,t}(\cdot \mid \mathbf{x}, \rho = 1)) \\ &= \sum_{\mathbf{z}} q_\psi(\mathbf{z} \mid \mathbf{x}) \Big[ \log q_\psi(\mathbf{z} \mid \mathbf{x}) - \log p_\theta(\mathbf{z} \mid \mathbf{x}) - \log \kappa_t(\mathbf{x}, \mathbf{z}) + \log Z_{\theta,t}(\mathbf{x}) \Big] \\ &= \mathbb{D}_{\mathrm{KL}}(q_\psi(\cdot \mid \mathbf{x}) \,\|\, p_\theta(\cdot \mid \mathbf{x})) - \mathbb{E}_{\mathbf{z} \sim q_\psi(\cdot \mid \mathbf{x})}[\log \kappa_t(\mathbf{x}, \mathbf{z})] + \log Z_{\theta,t}(\mathbf{x}). \end{aligned} \tag{49}$$

Here, $\log Z_{\theta,t}(\mathbf{x})$ is constant with respect to the candidate distribution $q_\psi$, because $\kappa_t$ is computed from the E-step proposal $q_{\psi^{(t)}}$ and is fixed during the M-step.

Rearranging Eqn. (49), we obtain

$$\begin{aligned} &\mathbb{E}_{\mathbf{z} \sim q_\psi(\cdot \mid \mathbf{x})}[\log \kappa_t(\mathbf{x}, \mathbf{z})] - \mathbb{D}_{\mathrm{KL}}(q_\psi(\cdot \mid \mathbf{x}) \,\|\, p_\theta(\cdot \mid \mathbf{x})) \\ &= \log Z_{\theta,t}(\mathbf{x}) - \mathbb{D}_{\mathrm{KL}}(q_\psi(\cdot \mid \mathbf{x}) \,\|\, p_{\theta,t}(\cdot \mid \mathbf{x}, \rho = 1)), \end{aligned} \tag{50}$$

which proves the variational identity in Eqn. (44).

Since KL divergence is nonnegative and equals zero if and only if its two arguments are identical almost surely,

$$\mathbb{D}_{\mathrm{KL}}(q_\psi(\cdot \mid \mathbf{x}) \,\|\, p_{\theta,t}(\cdot \mid \mathbf{x}, \rho = 1)) \geq 0, \tag{51}$$

with equality if and only if

$$q_\psi(\cdot \mid \mathbf{x}) = p_{\theta,t}(\cdot \mid \mathbf{x}, \rho = 1) \quad \text{a.s.} \tag{52}$$

Therefore,

$$\mathcal{L}_{\mathrm{VSD}}^{(t)}(q_\psi; \mathbf{x}) \leq \log Z_{\theta,t}(\mathbf{x}), \tag{53}$$

and the upper bound is uniquely achieved at

$$q_t^\star(\cdot \mid \mathbf{x}) = p_{\theta,t}(\cdot \mid \mathbf{x}, \rho = 1). \tag{54}$$

$\square$

# D. Extended Experiments

## D.1. Results for Multimodal Large Language Models

*Table 7.* Comparison of speedup ratio $SR$ and acceptance length $\tau$ on standard MLLM benchmarks with temperature $T \in \{0, 1\}$. The subscripts denote the relative improvement compared to the corresponding baseline. For example, at $T = 0$, MSD+VSD on LLaVA-1.5 7B achieves the average $SR$ of 2.45 with an additional +7.9% gain over the MSD value of 2.27.

| | Model | Method | VQAv2 | | AI2D | | SQA Image | | ChartQA | | TextVQA | | Hallusion | | Average | |
|---|---|---|---|---|---|---|---|---|---|---|---|---|---|---|---|---|
| | | | $SR\uparrow$ | $\tau\uparrow$ | $SR\uparrow$ | $\tau\uparrow$ | $SR\uparrow$ | $\tau\uparrow$ | $SR\uparrow$ | $\tau\uparrow$ | $SR\uparrow$ | $\tau\uparrow$ | $SR\uparrow$ | $\tau\uparrow$ | $SR\uparrow$ | $\tau\uparrow$ |
| Temperature = 0 | LLaVA-1.5 7B | Lookahead | 1.43 | 1.35 | 1.60 | 1.46 | 1.51 | 1.45 | 1.33 | 1.43 | 1.32 | 1.31 | 1.59 | 1.51 | 1.46 | 1.42 |
| | | Medusa | 1.63 | 1.52 | 1.71 | 1.69 | 1.59 | 1.57 | 1.50 | 1.62 | 1.49 | 1.51 | 1.74 | 1.99 | 1.61 | 1.65 |
| | | EAGLE-2 | 2.37 | 4.53 | 1.98 | 3.35 | 1.79 | 3.41 | 1.81 | 3.25 | 1.84 | 3.62 | 2.04 | 3.55 | 1.97 | 3.62 |
| | | MSD | 2.47 | 4.99 | 2.23 | 4.20 | 2.08 | 4.24 | 2.21 | 4.14 | 2.12 | 4.18 | 2.51 | 4.60 | 2.27 | 4.40 |
| | | MSD+VSD | 2.69 | 5.10 | 2.39 | 4.48 | 2.22 | 4.53 | 2.46 | 4.58 | 2.29 | 4.41 | 2.67 | 4.93 | 2.45$_{+7.9\%}$ | 4.67$_{+6.3\%}$ |
| | | **MSD+VSD (KL)** | **2.75** | **5.13** | **2.40** | **4.49** | **2.23** | **4.53** | **2.48** | **4.60** | **2.31** | **4.44** | **2.70** | **4.99** | **2.48$_{+9.0\%}$** | **4.70$_{+6.8\%}$** |
| | LLaVA-1.5 13B | Lookahead | 1.42 | 1.30 | 1.32 | 1.40 | 1.32 | 1.41 | 1.66 | 1.46 | 1.38 | 1.33 | 1.43 | 1.47 | 1.42 | 1.40 |
| | | Medusa | 1.50 | 1.41 | 1.47 | 1.61 | 1.39 | 1.50 | 1.75 | 1.56 | 1.51 | 1.56 | 1.51 | 1.58 | 1.54 | 1.61 |
| | | EAGLE-2 | 2.36 | 4.29 | 2.04 | 3.38 | 1.76 | 3.16 | 2.21 | 3.18 | 1.81 | 3.35 | 2.09 | 3.41 | 2.05 | 3.46 |
| | | MSD | 2.44 | 4.48 | 2.35 | 4.08 | 2.10 | 3.93 | 2.87 | 3.97 | 2.25 | 4.07 | 2.53 | 4.35 | 2.43 | 4.15 |
| | | MSD+VSD | 2.65 | 4.95 | 2.60 | 4.50 | 2.30 | 4.56 | 3.21 | 4.50 | 2.49 | 4.54 | 2.77 | 4.79 | 2.68$_{+10.1\%}$ | 4.64$_{+11.9\%}$ |
| | | **MSD+VSD (KL)** | **2.69** | **4.99** | **2.64** | **4.58** | **2.35** | **4.61** | **3.22** | **4.54** | **2.52** | **4.57** | **2.79** | **4.82** | **2.70$_{+11.1\%}$** | **4.69$_{+12.9\%}$** |
| Temperature = 1 | LLaVA-1.5 7B | Lookahead | 1.34 | 1.24 | 1.12 | 1.26 | 1.24 | 1.30 | 1.28 | 1.27 | 1.08 | 1.21 | 1.31 | 1.38 | 1.23 | 1.28 |
| | | Medusa | 1.51 | 1.93 | 1.19 | 1.43 | 1.38 | 1.87 | 1.36 | 1.82 | 1.12 | 1.43 | 1.47 | 1.88 | 1.34 | 1.73 |
| | | EAGLE-2 | 1.85 | 3.28 | 1.24 | 2.61 | 1.48 | 2.71 | 1.52 | 2.57 | 1.37 | 2.72 | 1.52 | 2.82 | 1.50 | 2.79 |
| | | MSD | 2.06 | 3.59 | 1.49 | 3.13 | 1.77 | 3.26 | 1.81 | 3.02 | 1.46 | 2.93 | 1.80 | 3.39 | 1.74 | 3.22 |
| | | MSD+VSD | 2.12 | 3.64 | 1.69 | 3.58 | 2.01 | 3.39 | 1.84 | 3.20 | 1.76 | 3.06 | 1.86 | 3.51 | 1.88$_{+8.2\%}$ | 3.40$_{+5.4\%}$ |
| | | **MSD+VSD (KL)** | **2.12** | **3.63** | **1.73** | **3.62** | **2.02** | **3.41** | **1.84** | **3.20** | **1.77** | **3.07** | **1.88** | **3.56** | **1.89$_{+9.0\%}$** | **3.42$_{+6.0\%}$** |
| | LLaVA-1.5 13B | Lookahead | 1.09 | 1.21 | 1.23 | 1.29 | 1.18 | 1.26 | 1.08 | 1.28 | 1.17 | 1.22 | 1.20 | 1.32 | 1.16 | 1.26 |
| | | Medusa | 1.16 | 1.35 | 1.31 | 1.43 | 1.25 | 1.39 | 1.16 | 1.87 | 1.21 | 1.94 | 1.35 | 1.42 | 1.24 | 1.57 |
| | | EAGLE-2 | 1.71 | 3.17 | 1.60 | 2.63 | 1.58 | 2.67 | 1.13 | 2.59 | 1.62 | 2.58 | 1.68 | 2.82 | 1.55 | 2.74 |
| | | MSD | 1.81 | 3.44 | 1.81 | 3.08 | 1.77 | 3.21 | 1.39 | 3.06 | 1.92 | 3.01 | 1.96 | 3.41 | 1.78 | 3.20 |
| | | MSD+VSD | 2.00 | 3.54 | 1.88 | 3.27 | 1.88 | 3.51 | 1.54 | 3.33 | 2.04 | 3.21 | 2.15 | 3.66 | 1.91$_{+7.8\%}$ | 3.42$_{+6.7\%}$ |
| | | **MSD+VSD (KL)** | **2.01** | **3.55** | **1.90** | **3.35** | **1.89** | **3.54** | **1.55** | **3.36** | **2.07** | **3.29** | **2.17** | **3.70** | **1.93$_{+8.7\%}$** | **3.47$_{+8.2\%}$** |

**Implementations.** We train speculative draft models based on the backbone of MSD (Lin et al., 2025) from scratch using the VSD objective in Eqn. (8). Following the token-level top-$K$ KL distillation loss design in HASS (Zhang et al., 2024), we replace the the KL penalty term $\mathbb{D}_{\mathrm{KL}}$ with the *top-10 renormalized KL divergence* $\mathcal{L}_{\mathrm{Top\text{-}10}}$ in Eqn. (56). Specifically, denote $p_\theta(\cdot)$ and $q_\psi(\cdot)$ as the next-token distributions of the target LLM and the draft model, respectively. Let $\Omega$ denote the full vocabulary, and let $\hat{\Omega} \subset \Omega$ be the set of the $K$ tokens with the highest probability under the target LLM $p_\theta(\cdot)$. We can define renormalized target and draft distributions based on the domain of $\hat{\Omega}$:

$$\hat{p}_\theta(x) \triangleq \frac{p_\theta(x)}{\sum_{y \in \hat{\Omega}} p_\theta(y)}, \quad \hat{q}_\psi(x) \triangleq \frac{q_\psi(x)}{\sum_{y \in \hat{\Omega}} q_\psi(y)}, \quad x \in \hat{\Omega}. \tag{55}$$

Then we adopt the following token-level top-$K$ distillation objective:

$$\mathcal{L}_{\mathrm{Top\text{-}K}} = -\sum_{x \in \hat{\Omega}} \hat{p}_\theta(x) \log \hat{q}_\psi(x). \tag{56}$$

In our setting, we use $K=10$ over $\hat{\Omega}$ to form the KL penalty used in VSD (KL). We report this VSD variant as VSD (KL) in Tab. 7.

The training dataset sizes for both text-only and visual instruction-tuning datasets are identical. Specifically, the text-only instruction-tuning dataset consists of 68,000 randomly selected dialogue samples from ShareGPT, while the visual instruction-tuning dataset includes 68,000 randomly selected samples from LLaVA-Instruct-150K. We train draft models on eight L-40S GPUs for 80 epochs, with a batch size of 64 and a learning rate of 3e-4. We adopt the AdamW optimizer with $\beta_1 = 0.9$, $\beta_2 = 0.95$, and zero weight decay. We use a cosine learning rate scheduler, in which the learning rate is linearly warmed up from 0 to $3 \times 10^{-4}$ over the first 2% of training steps, followed by cosine decay to a minimum learning rate of $0.01\times$ the peak value for the remainder of training. Gradient clipping is applied with a maximum norm of 0.5. We employ ZeRO Stage-2 optimization with overlapping communication enabled, using all-gather and reduce-scatter operations. The

communication bucket size is set to $2 \times 10^8$. Hyperparameters in VSD were tuned via a grid search. Across all experiments, we set sampling length $L = 7$ and sample size $S = 40$. We find that VSD is not sensitive to the bounded weighting function $1 < w(\ell) < 2, \forall l$. Thus, we fixed the weighting factor $w(\ell) := 1.05^l$, $\tau = 0.2$, and $\alpha = 0.1$, and VSD is generally robust to small variations in these hyperparameters. We adopt a dynamic draft tree with a fixed budget of 60 draft tokens, a maximum tree depth of 5, and a top-k of 8, following the configuration in prior works (Lin et al., 2025; Xie et al., 2025) and report results averaged over three independent runs on L-40S GPU.

**Results.** We report the acceptance lengths ($\tau$) and speedup ratios ($SR$) of VSD and VSD (KL) against all baselines across six benchmarks in Tab. 7. VSD and VSD (KL) consistently outperform all baselines across all datasets, models, and temperature settings.

As detailed in Tab. 7, our proposed VSD enhances inference efficiency across standard MLLM benchmarks, outperforming existing baselines. By applying VSD and VSD (KL) to MSD, we observe consistent gains in both speedup ratio ($SR$) and acceptance length ($\tau$) across varying model scales (LLaVA-1.5 7B and 13B) and sampling temperatures ($T \in \{0, 1\}$). **Case I: Greedy Decoding (T=0).** In the deterministic setting, VSD demonstrates robust generalization capabilities on LLaVA-1.5 13B, boosting the average speedup and acceptance length of MSD by up to **10.1%** and **11.9%**, respectively. Incorporating the top-10 KL term strengthens this effect: VSD (KL) yields up to **11.1%** $SR$ and **12.9%** $\tau$ improvements over MSD. **Case II:Stochastic Decoding (T=1).** In the stochastic sampling setting, VSD and VSD (KL) maintain superior performance; for LLaVA-1.5 13B, VSD improves MSD's average speedup by **7.8%** and acceptance length by **6.7%**, while VSD (KL) further improves these to **8.7%** and **8.2%**, respectively.

Thus, the results across diverse tasks, ranging from general visual recognition (e.g., VQAv2) to complex document reasoning (e.g., TextVQA), and models highlight the versatility and robustness of VSD. The consistent improvements, even at different temperatures, underscore VSD's effectiveness in handling varying levels of stochasticity in token predictions. In addition, the consistent improvement of VSD (KL) over VSD supports the role of the KL term in Eqn. (8): the KL regularizer explicitly encourages the draft distribution to align with the target distribution, which increases acceptance rate and thereby raises both $\tau$ and $SR$. Under this view, commonly used training objectives, such as standard cross-entropy (Li et al., 2026) or top-$K$ KL loss (Zhang et al., 2024), act as practical surrogates for the KL component of the variational objective; empirically, top-$K$ KL provides the strongest alignment signal and yields the best performance in our setting.

# E. Draft Tree Construction

We first revisit the draft tree construction in EAGLE-2 (Li et al., 2024b). Formally, given a training prefix (context) $\mathbf{x}$, EAGLE-2 constructs a depth-$d$ draft tree $\mathbf{T}_t$ with the draft model $M_q$:

$$\mathbf{T}_t = \mathcal{G}(M_q, \mathbf{x}), \tag{57}$$

where $\mathcal{G}$ denotes the decoding sampling policy. The decoding policy $\mathcal{G}$ grows the draft tree in two stages.

**(i) Layer-wise expansion phrase.** (see Eagle2 (Li et al., 2024b), Section 4.1)

At depth $\ell \in \{1, \ldots, d\}$, consider all frontier expansions (token edges) from the current layer. For each candidate expansion we compute a global acceptance score. We then select the top-$k$ token expansions across the entire layer according to the global acceptance score and expand draft tree only on these children. This global competition allows promising siblings to outcompete locally greedy choices and prevents early commitment to a single path.

**(ii) Global pruning and ranking phrase.** (see Eagle 2 (Li et al., 2024b), Section 4.2)

After reaching the maximum depth, we collect all leaves and rank them by the global acceptance score. We retain the top-$g$ leaves and prune the rest.

# F. Implementation Details

### F.1. Additional Experimental Details for Large Language Models

For EAGLE (Li et al., 2024a), EAGLE-2 (Li et al., 2024b), EAGLE-3 (Li et al., 2026), HASS (Zhang et al., 2024), GRIFFIN (Hu et al., 2025b), Medusa (Cai et al., 2024), and Hydra (Ankner et al., 2024), we directly utilized the publicly released draft model parameters provided by the respective authors. For methods that do not require draft model training, such as PLD, Lookahead, and SPS, we evaluated performance using official code from their GitHub repositories. We train

the draft models for 20 epochs on 8 RTX PRO 6000 GPUs, using a batch size of 64 and a base learning rate of $5 \times 10^{-6}$. We adopt the AdamW optimizer with $\beta_1 = 0.9$, $\beta_2 = 0.95$, and zero weight decay. The learning rate is linearly warmed up from 0 to $5 \times 10^{-6}$ during the first 2% of training steps and then linearly decayed over the remaining steps. Gradient clipping is applied with a maximum norm of 0.5. We employ ZeRO Stage-2 optimization with overlapping communication enabled, using all-gather and reduce-scatter operations. The communication bucket size is set to $2 \times 10^8$. Hyperparameters in VSD were tuned via a grid search. Across all experiments, we set sampling length $L = 7$ and sample size $S = 40$. We find that VSD is not sensitive to the bounded weighting function $1 < w(\ell) < 2, \forall l$. Thus, we fixed the weighting factor $w(\ell) := 1.05^l$, $\tau = 0.2$, and $\alpha = 0.1$, and VSD is generally robust to small variations in these hyperparameters. We adopt a standard dynamic draft tree with a fixed budget of 60 draft tokens, a maximum tree depth of 7, and a top-k of 10, following the configuration shown to be effective in EAGLE-3 (Li et al., 2026), and report results averaged over three independent runs on RTX PRO 6000 GPU.

### F.2. Experimental Settings for Multimodal Large Language Models

**Model & Tasks.** We assess the performance of VSD on widely used open-source LLaVA families (Liu et al., 2024b): LLaVA-1.5-7B and LLaVA-1.5-13B. We evaluate performance on six diverse multimodal reasoning benchmarks. The chosen benchmarks include visual question answering such as SQA Image (Lu et al., 2022), VQAv2 (Goyal et al., 2017), and AI2D (Kembhavi et al., 2016), document and chart understanding tasks including ChartQA (Masry et al., 2022) and TextVQA (Singh et al., 2019), multimodal hallucination evaluation benchmarks such as Hallusion (Guan et al., 2024). To ensure generalizability, we use consistent model weights across all tasks without task-specific fine-tuning. Following (Gagrani et al., 2024; Kang et al., 2025), we design prompts to elicit long, detailed responses with reasoning processes from the models.

**Baselines & Implementations.** MLLMs are designed to jointly process different modalities, allowing machines to interpret and generate content that combines different modalities, such as visual and textual inputs. Similar to LLMs, as MLLMs continue to grow in scale and complexity, their inference latency increases substantially, posing significant challenges for practical deployment. Applying speculative decoding to MLLMs is an emerging research direction. Specifically, recent works (Gagrani et al., 2024; Xie et al., 2025; Lin et al., 2025; Hu et al., 2025c; Huang et al., 2025; Jang et al., 2024) focus on adapting the speculative decoding methods in the domain of vision-language models (VLMs) by effectively fusing the visual tokens in the input features. However, they overlook the problem of the training-decoding misalignment in the draft model training.

We compare VSD against established speculative decoding frameworks originally designed for language models: Lookahead (Fu et al., 2024), Medusa (Cai et al., 2024), EAGLE-2 (Li et al., 2024b), and two SoTA VLM speculative decoding frameworks: MSD (Lin et al., 2025) and ViSpec (Kang et al., 2025). By default, VSD initializes its draft model from MSD and ViSpec. To adapt Lookahead, Medusa, and EAGLE-2 for VLMs, we modify their input pipelines to process image patch embeddings from the VLM's original vision encoder, enabling the draft models to generate speculative tokens conditioned on both visual and textual contexts. This adaptation is feasible as Lookahead, Medusa, and EAGLE-2 rely on general token prediction mechanisms that are theoretically compatible with multimodal sequences, provided the draft model can handle visual inputs. Vanilla autoregressive decoding serves as the baseline (speedup ratio = $1.00\times$). For a fair comparison, we do not relax the draft token acceptance condition of Medusa under non-greedy settings as proposed in the original paper; instead, we adopt the same acceptance condition as EAGLE-2.

We follow previous methods to train draft models. The training dataset sizes for both text-only and visual instruction-tuning datasets are identical. Specifically, the text-only instruction-tuning dataset consists of 68,000 randomly selected dialogue samples from ShareGPT, while the visual instruction-tuning dataset includes 68,000 randomly selected samples from LLaVA-Instruct-150K. We train draft models on eight L-40S GPUs for 40 epochs, with a batch size of 64 and a learning rate of 5e-6. We adopt the AdamW optimizer with $\beta_1 = 0.9$, $\beta_2 = 0.95$, and zero weight decay. We use a cosine learning rate scheduler, in which the learning rate is linearly warmed up from 0 to $5 \times 10^{-6}$ over the first 2% of training steps, followed by cosine decay to a minimum learning rate of $0.01\times$ the peak value for the remainder of training. Gradient clipping is applied with a maximum norm of 0.5. We employ ZeRO Stage-2 optimization with overlapping communication enabled, using all-gather and reduce-scatter operations. The communication bucket size is set to $2 \times 10^8$. Hyperparameters in VSD were tuned via a grid search. Across all experiments, we set sampling length $L = 7$ and sample size $S = 40$. We find that VSD is not sensitive to the bounded weighting function $1 < w(\ell) < 2, \forall l$. Thus, we fixed the weighting factor $w(\ell) := 1.05^l$, $\tau = 0.2$, and $\alpha = 0.1$, and VSD is generally robust to small variations in these hyperparameters. We adopt a dynamic draft tree with a fixed budget of 60 draft tokens, a maximum tree depth of 5, and a top-k of 8, following the configuration in prior

works (Lin et al., 2025; Xie et al., 2025) and report results averaged over three independent runs on L-40S GPU.

**Metrics.** For fairness and consistency, we follow priors and fix the decoding batch size to 1 and evaluate under decoding temperatures $T \in \{0, 1\}$. Same as prior works like EAGLE-2 (Li et al., 2024b), VSD is lossless and can preserve output quality. Thus, we focus on two efficiency metrics: (i) **Speedup Ratio** ($SR$) — the wall-clock time acceleration relative to vanilla decoding, and (ii) **Acceptance Length** ($\tau$) — the average number of tokens accepted per draft-verification cycle.

