# OpenReview forum: "Variational Speculative Decoding: Rethinking Draft Training from Token Likelihood to Sequence Acceptance"
_ICML.cc/2026/Conference — ICML 2026 regular_

### Official Review · Reviewer_X9FQ · 2026-03-07

**Soundness:** 3
**Presentation:** 4
**Significance:** 3
**Originality:** 3
**Overall Recommendation:** 5
**Confidence:** 3

**Summary:**

The paper argues that standard draft training for speculative decoding (token-level likelihood on a single path) is misaligned with the decoding objective, where efficiency depends on sequence-level acceptance under the target model. It proposes Variational Speculative Decoding (VSD): a variational objective that maximizes expected log acceptance of draft sequences while regularizing the draft distribution toward the target distribution. The authors optimize it with a sampling/EM-style procedure and add practical stabilizers (e.g., adaptive rejection weighting and confidence-aware regularization). Experiments on several LLM/MLLM settings show consistent improvements in acceptance length and speedup over strong baselines.

**Compliance With Llm Reviewing Policy:**

Affirmed.

**Key Questions For Authors:**

Please refer to the weaknesses above.

**Limitations:**

yes

**Strengths And Weaknesses:**

**Strengths**
- Clear motivation (optimizing acceptance at the sequence level).
- A principled variational formulation (with a connection to the expected accepted length).
- Consistent empirical improvements across multiple models/tasks.
- The paper is generally clear, and the training–decoding mismatch is well explained.

 **Weaknesses**
- The method is more complex, and training may be more costly. Compared to standard draft distillation, what is the end-to-end training cost of VSD (e.g., wall-clock time and GPU-hours)?
- Releasing experimental code would greatly help reproducibility.
- How sensitive are the results to key hyperparameters (e.g., sampling budget / number of proposals, ARW/CAR weights and schedules)?

---

> ### Author Rebuttal · Authors · 2026-03-31
>
> Thank you for your insightful comments, and also for your very careful proofreading! We provide our point-by-point response and hope our response helps address your concerns.
>
> **1.Regarding to the training framework,** we cast draft training as variational inference over latent proposals, derive an ELBO, and optimize it with EM. Algorithm 1 gives the procedure, and Appendix A.2 explains the rationale.
>
> **Practical Value.**
>
> We emphasize that the additional cost of VSD should be interpreted as a *one-time, amortized training investment* for high-throughput deployment scenarios, such as large-scale LLM services. In practical serving systems, even a modest improvement in decoding efficiency can accumulate into substantial system-level savings. Since VSD improves decoding efficiency *without sacrificing generation quality*, its training overhead can be justified by the deployment benefit below.
>
> **Amortization in deployment.**
> For large-scale services, an efficiency improvement of around 9\% can translate into tens of thousands of GPU-hours saved per day. This is substantially larger than the one-time training cost of VSD.
>
> **Loseless acceleration.**
> Another advantage of VSD is that it improves decoding efficiency while preserving generation quality, owing to the lossless nature of speculative decoding.
>
> **Training Cost Analysis.**
>
> VSD introduces additional cost mainly in the E-step, which generates latent proposals for the variational objective. The M-step adds negligible overhead, since it only reweights the standard training loss with scalar coefficients.
>
> With sampling length $L=7$ and sample size $S=40$, VSD requires about 440, 600, and 1300 GPU hours for the 8B, 13B, and 70B models on 8xRTX PRO 6000 GPUs. By comparison, training the EAGLE-3 model from scratch requires about 1000, 2000, and 5300 GPU hours. VSD adds only about 30\%, 23\%, and 20\% of draft-model training cost, respectively, while amortization in deployment and lossless acceleration can justify the relative overhead. The additional training cost is small relative to the recurring inference savings achieved after deployment.
>
> **Continual Training Analysis.**
>
> To address the concern that VSD may benefit simply from more optimization steps, we conducted an experiment to show its algorithmic advantage. For Llama-8B, we allocated the same additional training budget as VSD (440 GPU-hours) to the EAGLE-3 baseline, using its original training objective.
>
> |Method|MT-bench||HumanEval||GSM8K||Avg||
> |-|-|-|-|-|-|-|-|-|
> |**T=0**|SR|t|SR|t|SR|t|SR|t|
> |VSD|4.05|6.79|4.63|7.27|3.96|6.93|4.22|7.00|
> |EAGLE-3|3.79|6.31|4.29|6.87|3.77|6.47|3.95|6.55|
> |EAGLE-3 (continual training)|3.83|6.34|4.31|6.92|3.80|6.51|3.98|6.59|
> |**T=1**|SR|t|SR|t|SR|t|SR|t|
> |VSD|2.41|4.23|3.80|6.15|3.31|5.39|3.18|5.26|
> |EAGLE-3|2.32|4.10|3.48|5.86|3.10|5.07|2.97|5.01|
> |EAGLE-3 (continual training)|2.37|4.15|3.50|5.92|3.13|5.11|3.00|5.06|
>
> Even with the same training budget, EAGLE-3 performs almost identically to its continued-training checkpoint because it already reached saturated performance based on its training objective. At T=0 and 1, VSD achieves +5.9\% and +5.8\% average SR than EAGLE-3 (continual training), respectively. It confirms that VSD provides a specific algorithmic advantage for decoding efficiency. We will add the analysis in the revised manuscript.
>
> For MLLM experiments (Tab 2), VSD draft models were trained from scratch. VSD achieves up to 8.8\% improvement over ViSpec, demonstrating its superiority in decoding efficiency and validating the training overhead.
>
> **2.Regarding the reproducibility,** we provide a comprehensive Algorithm 1. Sec 5 and Appendix D specify all key hyperparameters to help reproducibility. We will release the model checkpoints and code.
>
> **3.Regarding the hyperparameters,** Table 4 demonstrates that VSD is robust across a wide range of settings while exhibiting a scaling law w.r.t. the sampling budget.
>
> **Latent Proposals ($S$).**
> Increasing $S$ consistently improves both the Speedup Ratio (SR) and the Acceptance Length (t).
>
> On average at T=0, increasing $S$ from 10 to 40 improves the SR from 3.97 to 4.06 and the acceptance length from 6.58 to 6.73. This confirms that VSD effectively leverages larger sampling budgets to better bridge the training-decoding misalignment.
>
> **ARW and CAR Mechanisms.**
> The Adaptive Rejection Weighting and Confidence-Aware Regularization are critical for managing gradient variance and path-level overconfidence.
>
> Incorporating ARW improves performance across all values of $S$. For example, at $S=40$ ($T=0$), adding ARW increases the average SR from 4.06 to 4.15. It confirms that ARW reduces variance in the stochastic updates, leading to a more efficient draft policy.
>
> Adding CAR on top of ARW provides a further boost, increasing the average SR from 4.15 to 4.22 at $S=40$ ($T=0$). CAR mitigates the influence of overconfident hard samples assigned high probability by the draft model, reducing verification overhead.

---

> > ### Author Rebuttal · Reviewer_X9FQ · 2026-04-01
> >
> > Thank you for the authors’ response. I have raised my score to 5. I hope the authors will consider open-sourcing their work in the future, as it would be beneficial to the community.

---

> > > ### Author Response · Authors · 2026-04-02
> > >
> > > We sincerely thank you for your detailed feedback and for considering our responses. We appreciate your constructive comment, and will release the model checkpoints and code.

---

### Official Review · Reviewer_Ekmx · 2026-03-12

**Soundness:** 2
**Presentation:** 3
**Significance:** 3
**Originality:** 2
**Overall Recommendation:** 4
**Confidence:** 3

**Summary:**

The authors investigate a notable topic: optimizing the draft model in speculative decoding to better align with the multi-path verification behavior of the target model. This article presents a central concept called Variational Speculative Decoding (VSD), which reformulates draft model training as a variational inference problem over latent draft paths. Instead of relying on token-level cross-entropy against a single greedy trajectory, VSD utilizes an EM-based MCMC framework where the draft model generates multiple candidate paths, the target model acts as an oracle to evaluate their acceptance probability, and the draft model is updated using adaptive rejection weighting and confidence-aware regularization to maximize the likelihood of accepted sequences.

**Compliance With Llm Reviewing Policy:**

Affirmed.

**Key Questions For Authors:**

1. What exact decoding parameters (e.g., pure sampling, top-p, top-k, temperature) were used to generate the $S=40$ latent proposals during training?
2. If truncation methods (like top-p) were applied during training to ensure proposal quality, how does this empirical deviation from the true distribution affect the theoretical optimality guarantees provided in Theorems 1 and 2?

**Limitations:**

Yes.

**Strengths And Weaknesses:**

Strengths
1. The perspective of shifting draft model training from token-level greedy matching to path-level distribution alignment using accepted trajectories is highly relevant and insightful for the speculative decoding community.
2. The empirical results demonstrate consistent improvements in acceptance length and wall-clock speedup across multiple large language models and multimodal models on diverse benchmarks.

Weaknesses:
1. The experimental setup introduces an unfair comparison by initializing the proposed method from fully converged official baseline checkpoints (e.g., EAGLE-3) and training for additional epochs, rather than evaluating all methods under an equal training budget from scratch.
2. The paper entirely omits quantitative metrics regarding the training wall-clock time and memory consumption, which is a critical flaw given the massive computational overhead of keeping the large target model in the training loop to verify 40 latent proposals per step.
3. The proposed variational objective and its gradient estimator essentially mirror Reinforcement Learning and Direct Preference Optimization (DPO) by explicitly contrasting accepted (positive) and rejected (negative) paths, yet the paper lacks meaningful discussion or empirical baseline comparisons against these established alignment paradigms.

---

> ### Author Rebuttal · Authors · 2026-03-31
>
> Thank you for the insightful and valuable comments! In the following, we provide our point-by-point response and hope our response helps address your concerns.
>
> **1.Regarding the experimental setup and overhead,** we clarify that the performance gains of VSD are not merely a result of extended training time, but the algorithmic advantages of our variational formulation.
>
> **Training Cost Analysis.**
>
> VSD introduces additional cost mainly in the E-step, which generates latent proposals for the variational objective. The M-step adds negligible overhead, since it only reweighs the standard training loss with scalar coefficients.
>
> With sampling length $L=7$ and size $S=40$, VSD requires about 440, 600, and 1300 GPU hours for the 8B, 13B, and 70B models on 8xRTX PRO 6000 GPUs. Training EAGLE3 model from scratch requires about 1000, 2000, and 5300 GPU hours. VSD adds only about 30\%, 23\%, and 20\% of draft-model training cost, respectively, while amortization in deployment and lossless acceleration below can justify the overhead. The additional overhead is small relative to the recurring inference savings after deployment.
>
> **Continual Training Analysis.**
>
> To address the concern that VSD may benefit simply from more optimization steps, we conduct a controlled experiment to show its algorithmic advantage. For Llama-8B, we allocate the same additional training budget as VSD (440 GPU-hours) to EAGLE3, using its training objective.
>
> |Method|MT-bench||HumanEval||GSM8K||Avg||
> |-|-|-|-|-|-|-|-|-|
> |**T=0**|SR|t|SR|t|SR|t|SR|t|
> |VSD|4.05|6.79|4.63|7.27|3.96|6.93|4.22|7.00|
> |EAGLE3|3.79|6.31|4.29|6.87|3.77|6.47|3.95|6.55|
> |EAGLE3 (continual training)|3.83|6.34|4.31|6.92|3.80|6.51|3.98|6.59|
> |**T=1**|SR|t|SR|t|SR|t|SR|t|
> |VSD|2.41|4.23|3.80|6.15|3.31|5.39|3.18|5.26|
> |EAGLE3|2.32|4.10|3.48|5.86|3.10|5.07|2.97|5.01|
> |EAGLE3 (continual training)|2.37|4.15|3.50|5.92|3.13|5.11|3.00|5.06|
>
> Even with the same training budget, EAGLE3 performs almost identically to its continued-training checkpoint because it already reached saturated performance based on its training objective. At T=0 and 1, VSD achieves +5.9% and +5.8% average SR than EAGLE3 (continual training), respectively. It confirms that VSD provides a specific algorithmic advantage for decoding efficiency. We will add the analysis in the revised manuscript.
>
> For the MLLM experiments (Tab 2), the VSD draft models were trained from scratch. VSD achieves up to 8.8\% improvement over ViSpec, demonstrating its effectiveness in optimizing decoding efficiency and validating the training cost.
>
> **Practical Value.**
>
> We emphasize that the additional cost of VSD should be interpreted as a *one-time, amortized training investment* for high-throughput deployment, such as large-scale LLM services. In practical systems, even a modest improvement in decoding efficiency can accumulate into substantial system-level savings.
>
> For large-scale services, an efficiency improvement of around 9\% can translate into tens of thousands of GPU-hours saved per day. This is substantially larger than the one-time training cost of VSD. Another advantage of VSD is that it improves decoding efficiency while preserving generation quality, owing to the lossless nature of speculative decoding.
>
> **2.We thank the reviewer for the insightful feedback regarding the relationship between VSD and RL.** Please refer to our response to the final question of ``Reviewer DaFV``.
>
> **3.Regarding latent proposal generation,** we use the top-10 token sampling and the sampling parameter is set as 0.8, which aligns with EAGLE and SGlang.
>
> Truncation does not change the ELBO objective (Eqn.7), so it validates our theoretical guarantees. In fact, it accelerates convergence toward the optimal draft distribution.
>
> Theorem 1: Optimizing the VSD objective increases the provable lower bound of the expected accepted length.
>
> Theorem 2: The unique maximizer of the objective is the valid-path posterior $p_{\theta}(z|x,\rho=1)$.
>
> Empirical evidence suggests that truncation acts as an informative prior, focusing the variational search on high-utility draft proposal regions. By filtering out low-probability noise in training, these methods help the draft distribution converge to the reference target distribution faster. We track the oracle pass rate of $S=40$ latent proposals during training based on our top-10 sampling and full distribution sampling.
>
> |Epoch|Pass Rate (VSD)|Pass Rate (Full Distribution)|
> |-|-|-|
> |0|38\%|38\%|
> |5|57\%|54\%|
> |10|69\%|67\%|
> |15|78\%|75\%|
> |20|83\%|81\%|
>
> The pass rates in both sampling methods are nearly identical, showing the draft policy effectively learns high-utility paths regardless of the sampling breadth. However, truncation is more efficient: our sampling method achieves a 8\% decrease in training wall-clock time and a 0.7\% increase in avg SR compared to the full distribution sampling. Truncation improves the practical wallclock efficiency of the training without sacrificing theoretical benefits.

---

> > ### Author Rebuttal · Reviewer_Ekmx · 2026-04-01
> >
> > Thank you for the authors’ response. I have raised my score to 4

---

> > > ### Author Response · Authors · 2026-04-02
> > >
> > > Thank you for your careful consideration and for revisiting your evaluation. We sincerely appreciate your constructive feedback, and we are glad that our revisions have satisfactorily addressed your concerns.

---

### Official Review · Reviewer_G48u · 2026-03-13

**Soundness:** 3
**Presentation:** 4
**Significance:** 3
**Originality:** 3
**Overall Recommendation:** 5
**Confidence:** 3

**Summary:**

This paper identifies a training–decoding mismatch in speculative decoding: while decoding operates over multiple sampled and ranked draft paths, training typically remains focused on token-level likelihood along a single trajectory. To address this, the paper proposes Variational Speculative Decoding (VSD), which trains the draft model with a path-level variational objective aligned with target-model acceptance. The method is technically well motivated, and the empirical results show consistent speedup improvements across both LLM and MLLM settings.

**Compliance With Llm Reviewing Policy:**

Affirmed.

**Final Justification:**

The rebuttal effectively addressed my main concerns and clarified the technical details, increasing my confidence in the soundness of the work. The paper presents a novel and meaningful direction for extending speculative decoding, with solid empirical support. Overall, the strengths outweigh the weaknesses, and the rebuttal positively changed my evaluation. I have increased my score to 5.

I also encourage the authors to consider open-sourcing their code to further benefit the community.

**Key Questions For Authors:**

- Can the authors clarify the practical training cost of VSD more explicitly, e.g., relative wall-clock/GPU overhead versus standard draft model fine-tuning, given the dependence on latent proposal sampling and the observed gains from larger $S$?
- How broadly do the authors expect the framework to apply beyond the learned-drafter settings evaluated here? The current results support compatibility with several drafter families, but it would be useful to understand whether similar acceptance-aware path-level ideas could extend to other speculative decoding variants or different acceptance-side designs.

**Limitations:**

See Weaknesses and Questions.

**Strengths And Weaknesses:**

### Strengths
- The paper addresses a real limitation of current speculative decoding methods, and the proposed path-level formulation is conceptually well motivated. The variational view is a meaningful step beyond standard token-level draft training.
- The empirical results are strong overall. VSD shows consistent improvements over competitive baselines across different tasks, decoding regimes, and model settings, and the ablations are helpful in supporting the full method.

### Weaknesses
- The practical training pipeline appears fairly heavy. Beyond the core variational objective, VSD relies on EM-style posterior approximation, proposal sampling with $S=40$, corrective resampling, and additional weighting/regularization components, while the paper does not quantify the training overhead very explicitly relative to standard drafter training.
- The empirical efficiency is convincing in the reported setting, but it is still somewhat narrow from a systems perspective: efficiency is evaluated at decoding batch size 1, and the experiments are tied to specific hardware setups, so it is unclear how stable the speedup gains are across different serving regimes or devices.

---

> ### Author Rebuttal · Authors · 2026-03-31
>
> Thank you for your insightful comments, and also for your very careful proofreading! In the following, we provide our point-by-point response and hope our response helps address your concerns.
>
> **1.We thank the reviewer for the insightful feedback regarding the training overhead.** We provide a detailed training cost analysis. Please refer to our response to the first question to ``Reviewer X9FQ``.
>
> **2.Regarding batch size and hardware,** we conduct experiments showing that VSD’s efficiency gains are robust.
>
> We evaluate batch sizes 4 and 8 on a single RTX PRO 6000 GPU using LLaMA-8B.
>
> |Method|BS|MT-bench||HumanEval||GSM8K||Avg||
> |-|-|-|-|-|-|-|-|-|-|
> |**T=0**||SR|t|SR|t|SR|t|SR|t|
> |EAGLE3|4|2.44|4.65|2.82|5.42|2.64|4.81|2.63|4.96|
> |VSD|4|2.66|5.02|3.06|5.74|2.77|5.16|2.83|5.31|
> |EAGLE3|8|1.54|4.68|1.80|5.44|1.68|4.80|1.67|4.97|
> |VSD|8|1.66|5.03|1.97|5.79|1.81|5.18|1.81|5.33|
> |**T=1**||SR|t|SR|t|SR|t|SR|t|
> |EAGLE3|4|2.04|3.12|2.58|4.71|2.30|3.89|2.30|3.91|
> |VSD|4|2.13|3.26|2.83|4.98|2.51|4.17|2.49|4.14|
> |EAGLE3|8|1.25|3.22|1.57|4.78|1.44|3.96|1.42|3.99|
> |VSD|8|1.35|3.30|1.69|5.01|1.58|4.22|1.54|4.17|
>
> Although speculative decoding generally yields smaller gains at larger batch sizes, VSD consistently outperforms EAGLE-3 by producing higher-quality draft trees that require fewer verification cycles. Averaged across tasks, VSD improves SR over EAGLE-3 by +7.8\% and +8.5\% for BS$=4,8$, respectively. This suggests that VSD better concentrates probability mass on promising branches, resulting in effective draft trees during decoding.
>
> We also evaluate VSD on an NVIDIA A100-80G GPU using LLaMA-8B.
>
> |Method|MT-bench||HumanEval||GSM8K||Avg||
> |-|-|-|-|-|-|-|-|-|
> |**T=0**|SR|t|SR|t|SR|t|SR|t|
> |EAGLE3|3.81|6.31|4.32|6.89|3.78|6.47|3.97|6.56|
> |VSD|4.08|6.79|4.67|7.30|3.98|6.95|4.25|7.01|
> |**T=1**|SR|t|SR|t|SR|t|SR|t|
> |EAGLE3|2.36|4.12|3.51|5.86|3.13|5.09|3.00|5.03|
> |VSD|2.44|4.26|3.82|6.17|3.32|5.42|3.20|5.28|
>
> VSD improves average SR over EAGLE-3 by +6.9\% at $T=0$ and +6.5\% at $T=1$, confirming that the speedup is stable across hardware. We will include these results in the final version.
>
> **3.Regarding the extension of VSD,** the idea is to shift from token-level marginal optimization to path-level joint optimization under a variational inference framework via the utility $\kappa(x,z)$. This perspective is not limited to learned drafters and can extend to other speculative decoding designs.
>
> **Extension to draft-model-free variants.**
>
> In variants like $n$-gram decoding (arXiv:2404.08698) or Lookahead decoding (arXiv:2402.02057), the proposal mechanism (e.g., $n$-gram modules or Jacobian trajectories) can be viewed as an implicit proposal distribution $q$. The principle beyond VSD can refine these methods by favoring token sequences with high joint acceptance probability, rather than token-level.
>
> N-gram decoding utilizes an adaptive module where the proposal probability is based on frequentist counts of historical token sequences:
> $$P(x_t|x_{t-N+1},\dots,x_{t-1})\approx\frac{\text{count}(x_{t-N+1},\dots,x_t)}{\text{count}(x_{t-N+1},\dots,x_{t-1})}$$
>
> We can extend this by redefining the update rule so that N-gram weights are scaled by their path-level validity $\kappa(x, z)$. The module is updated proportionally to the acceptance of the entire path $z$:
> $$\text{Weight}(x_{t-N+1},\dots,x_t)\leftarrow\text{Weight}+\eta\cdot\kappa(x, z)$$
> This transforms the frequentist cache into a variational sampler that prioritizes sequences with high joint acceptance probability. VSD also can extend N-gram modules to Block-level, as discussed below.
>
> Lookahead decoding reformulates autoregressive generation as solving a non-linear system through fixed-point Jacobi iteration. The Lookahead Branch generates multiple disjoint $n$-grams from the Jacobi trajectory, while the Verification Branch selects promising candidates from an $n$-gram pool based on prefix matching. The candidate scoring can be based on the expected path-level utility $\mathbb{E}\_{q_{\psi}}[\log\kappa(x, z)]$. To maintain decoding efficiency without additional overhead, a practical approximation of this probability is the log-likelihood of the draft model. We can use this score to estimate acceptance rates. This enables Lookahead to decide which Jacobi trajectories are most likely to converge to valid fixed points, prioritizing trajectories with high path-level utility.
>
> **Extension to Acceptance-side.**
>
> VSD’s path-level objective is also related to block-level verification (arXiv:2403.10444). In block-level verification (Algorithm 2), the key state variable is the cumulative probability ratio $p_i$, which represents the joint probability that a sub-block $X^i$ is accepted. VSD’s path-level utility $\kappa(x, z)$ is the exact training-side counterpart to this $p_i$. Thus, VSD improves the draft policy by optimizing path-level acceptance during training, while block-level verification improves the verification from the same joint-distribution perspective.

---

> > ### Author Rebuttal · Reviewer_G48u · 2026-04-02
> >
> > Thank you to the authors for the detailed and thoughtful responses. My concerns have been fully addressed, and I appreciate the clarifications and discussion provided. I believe this work has the potential to make a meaningful contribution to expanding the paradigm of speculative decoding, and I look forward to its impact on future research in this area. I will raise my score to 5. As a final suggestion, I encourage the authors to consider open-sourcing their code, as it would further benefit the community and enhance the impact of this work.

---

> > > ### Author Response · Authors · 2026-04-02
> > >
> > > We sincerely thank the reviewer for the constructive comments and suggestions, which are very helpful for improving our paper. We will release the model checkpoints and code.

---

### Official Review · Reviewer_DaFV · 2026-03-16

**Soundness:** 3
**Presentation:** 3
**Significance:** 2
**Originality:** 3
**Overall Recommendation:** 4
**Confidence:** 4

**Summary:**

This paper identifies a training-decoding distributional discrepancy in speculative decoding: draft models are trained via token-level cross-entropy on single greedy trajectories, yet at inference they must produce multiple stochastic draft paths that are ranked and verified at the path level. The authors propose Variational Speculative Decoding (VSD), which reformulates draft training as variational inference over latent draft paths, deriving an ELBO that encourages the draft model to place probability mass on paths likely to be fully accepted by the target model while staying close to the target distribution via a KL regularizer. Optimization proceeds via an EM algorithm: the E-step draws MCMC samples filtered by an oracle that compares path-level validity to a greedy baseline, and the M-step updates the draft model using a variance-reduced gradient estimator incorporating Adaptive Rejection Weighting (ARW) and Confidence-Aware Regularization (CAR). Theoretical results link the VSD objective to a lower bound on expected acceptance length. Experiments on four LLMs (LLaMA-3.1-8B, LLaMA-3.3-70B, Vicuna-13B, DeepSeek-R1-8B) and two MLLMs (LLaVA-1.5-7B/13B) show consistent improvements in acceptance length and wall-clock speedup over EAGLE-3, MSD, and ViSpec baselines.

**Compliance With Llm Reviewing Policy:**

Affirmed.

**Final Justification:**

The rebuttal adequately addressed my primary concerns, particularly regarding training overhead, fairness of the comparison, and the relationship to RL-based alternatives. The remaining weaknesses are better viewed as limitations of scope and system complexity rather than fundamental flaws in the method. Therefore, I keep my original assessment.

**Key Questions For Authors:**

1. **What is the training cost overhead of VSD relative to standard cross-entropy training (e.g., EAGLE-3's training pipeline)?** Specifically, how many additional target model forward passes are required, and what is the wall-clock training time ratio?
   - **Why it matters:** If VSD doubles or triples training time, its 7–10% inference speedup improvement needs to be weighed against amortization over deployment scale.
   - **Effect on my evaluation:** A clear cost analysis showing reasonable overhead would increase my score; evidence of very high overhead without discussion of amortization would decrease it.

2. **How does the oracle acceptance rate (fraction of draft proposals passing the oracle filter in Algorithm 2) evolve during training, and how sensitive is performance to the choice of baseline in the oracle (greedy target vs. other baselines)?**
   - **Why it matters:** If the oracle is too selective, most training signal comes from target-model samples rather than draft model exploration, potentially limiting learning. If too permissive, the path-level filtering loses its purpose.
   - **Effect on my evaluation:** Evidence of well-calibrated oracle behavior would address my concern about the oracle design; evidence of extreme selectivity would weaken the contribution.

3. **Can the authors provide a direct comparison with an RL-based approach (e.g., REINFORCE with acceptance length as reward) for fine-tuning draft models?**
   - **Why it matters:** This is arguably the most natural alternative approach to the same problem, and its absence makes it difficult to assess whether the variational formulation provides genuine advantages beyond reframing.
   - **Effect on my evaluation:** A favorable comparison would strongly support the paper's methodological contribution; an unfavorable comparison or lack thereof is a notable gap.

**Limitations:**

The training cost discussion should be more explicit, and the sensitivity to oracle design choices deserves attention.

**Strengths And Weaknesses:**

## Strengths

1. **Well-motivated problem formulation.** The paper provides compelling empirical evidence (Figure 1) that the greedy path matches the accepted path only ~36% of the time and that the greedy path yields substantially shorter acceptance lengths than alternative high-confidence candidates. This concretely demonstrates that standard cross-entropy training is suboptimal for speculative decoding, providing genuine motivation for a path-level objective.

2. **Principled variational framework with clean theoretical grounding.** The formulation of draft training as variational inference over latent proposals is elegant. The ELBO decomposition into a path-acceptance term and a KL regularizer is natural and well-justified. Theorem 1 (VSD objective lower-bounds expected acceptance length) and Theorem 2 (the optimal draft distribution equals the valid-path posterior) provide a rigorous theoretical foundation. The connection between the variational objective and wall-clock speedup is particularly useful for practitioners.

3. **Extensive and convincing experimental evaluation.** The paper evaluates across four LLMs and two MLLMs, three text benchmarks, six multimodal benchmarks, two temperature settings, and multiple baseline methods (11 baselines in the LLM setting). The compatibility experiments (Table 3) showing VSD improves GRIFFIN and HASS draft models in addition to EAGLE-3 demonstrate that VSD is genuinely a complementary training framework rather than being tied to a specific architecture. The ablation study (Table 4) systematically validates ARW, CAR, and the scaling behavior of the number of proposals S.

---

## Weaknesses

1. **Training cost is insufficiently analyzed.** The paper does not report the additional training time or compute cost introduced by VSD. The E-step requires generating S proposals per prefix per length, evaluating the oracle (which requires target model forward passes to compute path-level validity), and the M-step involves additional gradient computation with ARW/CAR. For S=40 proposals and L sampling lengths, this could multiply training cost substantially relative to standard cross-entropy training. Without a training cost vs. inference speedup trade-off analysis, it is difficult to assess the practical value proposition. The authors note they could not scale beyond S=40 due to "computational constraints," which itself suggests non-trivial overhead.

2. **The oracle design introduces a potentially problematic bias.** Algorithm 2 reveals that the oracle compares each proposal's log path-validity to that of the greedy target path, returning a binary indicator weighted by w(l). This means the oracle only accepts proposals that are better than the greedy target baseline — a potentially very selective filter, especially early in training when the draft model is weak. The paper does not analyze how often proposals pass the oracle filter across training, the sensitivity to the choice of baseline (why greedy target specifically?), or how the weighting factor w(l) is set. This design choice seems underspecified and could substantially affect training dynamics.

3. **Limited analysis of when and why VSD helps most.** While improvements are consistent, they vary considerably across benchmarks and models (e.g., 12.6% SR improvement for Vicuna-13B at T=0 vs. 5.7% for LLaMA-3.3-70B at T=0). The paper does not analyze what properties of the task or model determine the magnitude of improvement. Understanding whether VSD helps more when the draft-target distribution gap is larger, when the vocabulary is more structured, or when sequences are longer would significantly strengthen the contribution. The T=0 vs T=1 gap (VSD helps more at T=0) is also unexplained beyond noting the numbers.

4. **The "MCMC" framing is somewhat loose.** The E-step is described as "oracle-filtered MCMC sampling," but it is really rejection sampling with replacement from the target model when proposals fail. There is no Markov chain in the classical MCMC sense (no transition kernel, no mixing analysis, no convergence guarantees for the approximate posterior). While this does not invalidate the algorithm, the terminology is imprecise and may mislead readers about the nature of the approximation being made. Standard variational inference literature would call this importance-weighted or rejection-based posterior approximation.

5. **Missing comparison with reinforcement learning approaches.** The training-decoding discrepancy the paper identifies could also be addressed by treating draft training as an RL problem where the reward is acceptance length. The paper does not discuss or compare against RL-based fine-tuning of draft models (e.g., using REINFORCE or PPO with acceptance length as reward), which would be a natural baseline given the stated problem.

---

> ### Author Rebuttal · Authors · 2026-03-31
>
> Thank you for the insightful and valuable comments! In the following, we provide our point-by-point response and hope our response helps address your concerns.
>
> **1.We thank the reviewer for the constructive feedback regarding the training overhead.** We provide a detailed training overhead analysis. Please refer to our response to the first question to ``Reviewer X9FQ``.
>
> Regarding "computational constraints," we clarify that our $S=40$ cap was driven by GPU VRAM limits (hold mini-batches of latent proposals and KV cache) rather than a time-based bottleneck.
>
> **2.Regarding the oracle design,** the oracle in Algorithm2 is a principled way to maximize the path-level validity term $\mathbb{E}\_{q_{\psi}(z|x)}[\log \kappa(x,z)]$ in the ELBO. It encourages the draft model to focus on trajectories whose utility matches or exceeds a reference. We use the greedy target path as the reference because it represents the standard upper bound in traditional distillation.
>
> We track the oracle pass rate during VSD training on LLaMA-8B ($S=40, L=7$), and compare the greedy-path oracle (O1) with a moving-average oracle (O2) that accepts proposals above the batch mean log-validity.
>
> |Epoch|Pass Rate (O1)|Pass Rate (O2)|
> |-|-|-|
> |0|38\%|38\%|
> |5|57\%|49\%|
> |10|69\%|57\%|
> |15|78\%|61\%|
> |20|83\%|63\%|
>
> The greedy oracle pass rate rises to 83\%, showing that the draft model increasingly proposes high-utility paths that surpass the greedy reference.
>
> |Oracle Method|Training Time (GPU hours)|Avg SR|Avg t|
> |-|-|-|-|
> |O1|440|3.70|6.13|
> |O2|524|3.62|5.93|
>
> Greedy-path oracle reduces training time by 15.3\% and improves average SR by 2.3\% than moving average oracle across MT-bench, HumanEval, GSM8K, indicating faster convergence and better performance.
>
> We set $w(l)=1.05^l$. VSD is robust to the choice as long as $w(l)$ remains bounded ($1<w(l)<2$), suggesting that the main training signal comes from the path-level utility (Appendix D).
>
> **3.Regarding the result analysis,** we conclude:
>
> VSD tends to yield larger gains when the draft-target gap in standard greedy training is larger. When greedy training collapses to suboptimal paths that deviate from the target model’s preferred trajectories, VSD provides greater benefit. Our EAGLE-3 experiments show that tasks with lower Greedy Path Acceptance Rate consistently obtain larger improvements from VSD. This explains why Vicuna-13B achieves a 13.6\% SR gain at $T=0$, while LLaMA-70B gains 5.7\%: Vicuna-13B has a lower average Greedy Path Acceptance Rate (30\%) than LLaMA-70B (36\%), indicating that the stronger model’s greedy paths are already better aligned and leave less room for VSD to improve.
>
> VSD also shows larger gains on tasks requiring long-horizon consistency. Benchmarks with longer average outputs, such as MT-Bench and HumanEval for language, and ChartQA and AI2D for vision, exhibit the largest improvements across model sizes.
>
> VSD is more effective at T=0 because the target distribution is more concentrated, making the valid-path posterior sharper. Thus, aligning the draft policy to the path-level posterior brings a larger advantage. At T=1, the target distribution has higher entropy and accepts a wider range of continuations, so the benefit of improved efficiency is smaller under the dispersed valid-path posterior.
>
> **4.Regarding MCMC,** we agree that our method is better described as a Monte Carlo stochastic posterior approximation. We will replace MCMC-based EM with Monte Carlo-based Variational EM.
>
> **5.Regarding the relationship between VSD and RL,** draft training can be cast as an RL problem with acceptance length as the reward. We compare VSD with two RL baselines: REINFORCE with a KL penalty and DPO.
>
> For REINFORCE, we treat the draft model $q_\psi$ as the policy, sample $S$ draft paths, and use the verifier’s acceptance length as the reward, with a KL penalty to prevent collapse. For DPO, we construct $S$ preference pairs from sampled draft paths during training, where the preferred path has a longer acceptance length. We evaluate all methods on LLaMA-8B under a 100 GPU-hour budget with $S=10$ and $\beta=0.1$ across MT-bench, HumanEval, GSM8K.
>
> |Method|Avg SR|Avg t|
> |-|-|-|
> |**T=0**|||
> |VSD|4.08|6.80|
> |REINFORCE |3.96|6.57|
> |DPO|3.99|6.58|
> |**T=1**|||
> |VSD|3.06|5.17|
> |REINFORCE|2.98|5.03|
> |DPO|3.00|5.04|
>
> VSD outperforms both RL baselines, improving average SR over REINFORCE by 3.0\% at T=0 and 2.6\% at T=1. Compared with standard RL, VSD has several advantages. 1) It is tied to inference efficiency: the variational objective has a closed-form connection to expected acceptance length. 2) VSD has a desirable target distribution: its optimum is the valid-path posterior $p_{\theta}(z\mid x,\rho=1)$, so training becomes posterior alignment rather than heuristic reward maximization. 3) VSD is more stable: ARW and CAR reduce variance, whereas policy-gradient methods often suffer from noisy updates under sparse rewards. We will include these analysis in our final version.

---

> > ### Author Rebuttal · Reviewer_DaFV · 2026-04-02
> >
> > Thank you for the detailed rebuttal. The additional results substantially improve my confidence in the paper: the training-cost analysis is now much clearer, the oracle design is better justified through pass-rate and alternative-baseline experiments, and the RL comparisons help position the variational formulation against a natural competing approach. The clarification on terminology is also appreciated. While I still view the method as somewhat complex from a training-systems perspective, I find the paper meaningfully stronger after rebuttal and remain positive on acceptance

---

> > > ### Author Response · Authors · 2026-04-02
> > >
> > > We sincerely appreciate your time and constructive feedback. Your insightful comments and suggestions have been very helpful in strengthening our manuscript.

---

### Decision · Program_Chairs · 2026-04-30

**Decision:**

Accept (regular)

**Comment:**

This paper improves speculative decoding by addressing the mismatch between standard token-level, single-path draft training and the multi-path, path-level behavior that determines decoding efficiency at inference time. The proposed method, Variational Speculative Decoding (VSD) demonstrates strong empirically gains. The reviewers broadly agreed on the novelty of shifting from token-level training to sequence-level acceptance, and the significance of the empirical gains.

The strong empirical gain of the method is a clear strength. The paper reports consistent improvements in acceptance length and wall-clock speedup across a broad set of models and benchmarks, and the compatibility and ablation experiments shows that VSD is not tied to one specific drafter. The rebuttal also addressed several important practical concerns raised in the initial reviews, including training overhead, equal-budget comparisons against continued baseline training, comparisons to RL-style alternatives, and robustness across batch sizes and hardware. All reviewers mentioned that their concerns are fully resolved after the rebuttal.

That said, I have a remaining concern about the theoretical framing, in particular the presentation of Theorem 2. As written, the quantity on the right-hand side of Eq. (13) depends on the draft distribution itself since $\kappa$ depends on the draft distribution. This makes the claim a bit weird since the optimality is with respect to a changing target. One interpretation is that the claim is a fixe-point type of claim but it doesn't seem to be the case based on the proof. I communicated this to the reviewers during the rebuttal and one review mentioned that this presentation also confused them. This does not change the empirical contribution, but it does make the current “principled” framing feel somewhat overstated. I believe the paper should be more careful in how strongly it presents the method as theoretically grounded.

Overall, however, I do not think this issue is sufficient to overturn the paper’s overall positive case. Given the generally positive reviewer consensus and the rebuttal’s substantial clarification of the practical concerns, I would recommend a weak acceptance. In the final version, I would strongly encourage the authors to revise the presentation of Theorem 2 and related claims, explicitly clarify any dependence on the draft distribution, and modify the surrounding language to avoid the overstatement about the connection between the theory and empirical demonstration.